

# Mapping snow depth in open alpine terrain from stereo satellite imagery

R. Marti[1,2], S. Gascoin[2], E. Berthier[3], M. de Pinel[4], T. Houet[1], and D. Laffly[1]

[1]Géographie de l'Environnement (GEODE), UT2J/CNRS, Toulouse, France
[2]Centre d'Etudes Spatiales de la Biosphère (CESBIO), UPS/CNRS/IRD/CNES, Toulouse, France
[3]Laboratoire d'Etudes en Géophysique et Océanographie Spatiales, (LEGOS), UPS/CNRS/IRD/CNES, Toulouse, France
[4]GeoFalco, Longages, France

*Correspondence to:* S. Gascoin (simon.gascoin@cesbio.cnes.fr)

**Abstract.** To date, there is no direct approach to map snow depth in mountainous areas from spaceborne sensors. Here, we examine the potential of very-high-resolution (VHR) stereo satellites to this purpose. Two triplets of 70 cm-resolution images were acquired by the Pléiades satellite over an open alpine catchment (14.5 km$^2$) under snow-free and snow-covered conditions. The open-source software Ame's Stereo Pipeline (ASP) was used to match the stereo pairs without ground control points, to generate raw photogrammetric clouds and to convert them into high-resolution Digital Elevation Models (DEMs) at 1-m, 2-m, and 4-m resolutions. The DEMs difference (dDEM) were computed after 3D-coregistration, including a correction of a $-0.48$ m vertical bias. The bias-corrected dDEMs maps were compared to 451 snow probe measurements. The results show a decimetric accuracy and precision in the Pléiades-derived snow depths. The median of the residuals is $-0.16$ m, with a standard deviation (SD) of 0.58 m at a pixel size of 2 m. We compared the 2 m-Pléiades dDEM to a 2 m-dDEM that was based on a winged unmanned aircraft vehicle (UAV) photogrammetric survey that was performed on the same winter date over a portion of the catchment (3.1 km$^2$). The UAV-derived snow depth map exhibit the same patterns as the Pléiades-derived snow map. The Pléiades images benefit from a very broad radiometric range (12 bits), allowing a high correlation success rate over the snow-covered areas. This study demonstrates the value of VHR stereo satellite imagery to map snow depth in remote mountainous areas without any field data.

## 1 Introduction

The seasonal snow cover in mountainous areas is important for hydropower production, irrigation, urban supply, risk assessment and recreation (Barnett et al., 2005). Snow cover sustains mountain glaciers, alters frozen ground through its insulating effect, and plays a major role in mountainous ecosystems and plant survival (Keller et al., 2005). The seasonal snow on the ground can be characterized by various metrics, including the snow covered area (SCA), the snow height (HS), the snow density $\rho_s$, and the snow equivalent in water (SWE) (Fierz et al., 2009). A key moment to evaluate the snow cover as a water resource in an alpine catchment is the accumulation peak, when the SWE reaches its maximum value. Even for small mountain catchments with areas of a few square kilometres, the spatial variability of the snow height and water equivalent is high because of the large ranges in elevation, aspect and land cover types (Grünewald et al., 2010).



Various techniques exist to monitor the HS and SWE at specific locations. The snow course is a standard protocol that is used to measure the SWE in the catchment areas of dams in many countries (DeWalle and Rango, 2008). An operator measures the HS with a snow probe at a number of predefined waypoints. The survey is repeated a few times during winter to obtain the amount of accumulated snow before spring freshets. The snow density is also estimated during a snow course, but this measurement is not conducted at every point because coring and weighing the snowpack takes a longer time than snow depth measurements (Sturm et al., 2010). In addition, many studies showed that the snow density is much less variable in space than the snow depth (Pomeroy and Gray, 1995; Marchand and Killingtveit, 2005; Jonas et al., 2009; López-Moreno et al., 2013). The snow course remains a time-consuming task, which can be dangerous because of the risk of avalanches. Even in small catchments, this approach does not enable field operators to routinely sample the entire catchment area. Automatic measurements that are based on snow pillows, sonic rangers, and nuclear snow gauges are widely used in addition to manual measurements (Egli et al., 2009). GPS interferometry has been recently used to measure the HS at decimetre resolution (Larson et al., 2009; Gutmann et al., 2012) and could represent an alternative in snow-dominated regions, where geodetic GPS receivers are already operating for various purposes (e.g., plate deformation or weather monitoring). All these point-scale observations must be extrapolated by using statistical models and/or remotely-sensed data (e.g. Martinec and Rango, 1981; Luce et al., 1999; Molotch et al., 2005; López-Moreno and Nogués-Bravo, 2006; Grünewald et al., 2013).

Remote sensing techniques are particularly suitable for monitoring snowpacks at the catchment scale under satisfactory safety conditions. Recent advances in the fundamental understanding of the distribution of mountain snow depth have been achieved through airborne Lidar (Light detection and ranging) campaigns (Deems and Painter, 2006; Deems et al., 2013). Lidar provides an accurate measurement of the snow depth with a very high spatial resolution, which is perfectly suited for monitoring snowpacks in mountainous areas, including in forested areas (Hopkinson and Sitar, 2004; Grünewald et al., 2013). The vertical accuracy ranges from of centimetres to a few decimetres (Grunewald and Scheithauer, 2010; Deems et al., 2013). This technique is being extended for operational purposes in the USA (Painter and Berisford, 2014, Airborne Snow Observatory http://aso.jpl.nasa.gov/). However, airplane surveys are costly and do not allow global coverage. Terrestrial laser scanners (TLS) are relatively less expensive than Airborne Laser Scanner (ALS) and offer comparable resolution and accuracy at mid-range distances (up to 300-500 m) (Prokop, 2008; Grünewald et al., 2010). The beam divergence of TLS is generally lower over steep terrain, but coarser over flat areas, which highlights the complementary nature of both ALS and TLS techniques in mountainous terrain.

Airborne and terrestrial photogrammetry has been investigated on snow surfaces since the 1960s (Cooper, 1965; Smith et al., 1967; Otake, 1980; Cline, 1993, 1994). Nevertheless, their successful assessment has been achieved only recently (Ledwith and Lundén, 2001; Lee and Jones, 2008; Bühler et al., 2015; Nolan et al., 2015; Jagt et al., 2015). Airborne photogrammetry represents a relatively inexpensive alternative to Lidar to generate accurate and precise HS maps. However, its use implies the presence of an operator to drive an unmanned aircraft vehicle (UAV)(Jagt et al., 2015), or a pilot to fly an airplane (Bühler et al., 2015; Nolan et al., 2015).

Satellite snow cover observations, including operational applications, have been performed for many decades(e.g. Rango A, 1976, 1994; Dietz et al., 2012). Numerous satellite-derived products exist at the global scale (Frei et al., 2012). Snow





cover maps (SCA) are routinely produced from visible or near-visible bands (e.g., MODIS products (Hall et al., 2002)). When combined with a distributed snowmelt model, the SWE can be reconstructed from the monitoring of the SCA, provided that the last day of snow on the ground is known (e.g. Molotch and Margulis, 2008). An important limitation of this method for operational purposes is that it requires the user to wait until the end of the snow season.

Microwave remote sensing techniques have been demonstrated to be effective for monitoring snowpack-related metrics (SCA, HS, SWE, wet/dry state  Sokol et al., 2003). Numerous spaceborne radiometers with appropriate frequency channels have been in orbit since the 1960s (e.g., SMMR 1978; SSM/I 1987; AMSR-E 2002). However, the application of passive microwaves to snowpack monitoring in alpine regions is limited by the coarse resolution of spaceborne sensors, which are typically 10-25 km (Clifford, 2010), and the presence of liquid water in the snowpack. Another limitation is the SWE threshold,

which impedes SWE retrieval for deep snowpacks (>150 mm). Several attempts have been made to retrieve spatially-distributed HS or SWE data from space by radar imagery (Papa et al., 2002; Leinss et al., 2014; Rott et al., 2014; Dedieu et al., 2014). However, the optimal frequency channels (ku,ka) are still absent from current SAR satellites. Radar can operate even under cloudy conditions, but snow penetration from band X or band C complicates these measurements, and large areas may remain masked because of the oblique view of the imager.

Satellite altimetry (e.g., ICESat) could potentially accurately determine the snow depth, but the large footprint is not optimal for small alpine catchments. Errors may arise from signal saturation and beam penetration. To date, there is no direct approach to map snow depth in mountainous areas from spaceborne sensors (Lettenmaier et al., 2015).

The objective of this paper is to assess the potential of stereo images from a very-high resolution (VHR) satellite to retrieve the snow depth. Recently, DEMs that were derived from Pléiades satellites have been assessed over various types of surfaces,

such as end-of-summer glacier surfaces (Marti et al., 2014; Berthier et al., 2014), lake deposits and dunes (Schuster et al., 2014; Lucas et al., 2015), or landslide areas (Stumpf et al., 2014; Lacroix et al., 2015). Pléiades-derived DEMs exhibited sub-meter accuracy in the elevation of these rugged topographies, which opens the possibility to sense the snow depth from space by subtracting a DEM that was obtained under snow-free conditions from a DEM that was obtained near the peak of snow accumulation. This study's goals are as follows:

– Generate, co-register, and differentiate two Pléiades DEMs in a small mountainous catchment without ground control points: a snow-free DEM and a DEM that was acquired near the snow accumulation peak;

– Assess the quality and accuracy of the difference in the Pléiades DEMs (dDEMs) based on two datasets: (i) snow probe measurements and (ii) another dDEM that is generated from two UAV surveys;

– Discuss the influence of the topography and land cover on the residuals between the Pléiades dDEM and the snow probe

measurements.



## 2 Study site

The study area is the Bassiès catchment ($14.5\,\mathrm{km}^2$), which is an open alpine terrain in the north-eastern Pyrenees (Fig. 1). Bassiès is one of the main sub-basins of the Upper Vicdessos Valley, which has a long history of hydropower production (Taillefer, 1939; Antoine et al., 2012). The elevation ranges between 1156 and 2676 m a.s.l. (median elevation 1659 m) with a contrasted relief: while steep slopes delimit the watershed, the valley bottom is rather flat, and exhibits gentle slopes in its central part. The catchment is ungauged, but the streamflow at the outlet is diverted toward a hydropower plant operated by "Electricité de France". The average annual temperature in the area is 6.6 °C and the precipitation is 1640 mm, of which at least 30 % falls as snow (Szczypta et al., 2015). The snow season generally starts in November-December and ends in May-June (Fig. 2). The catchment is 65% covered by subalpine meadow and 25% by rock and bare soil. The last 10% is composed of intermediate vegetation (scattered short-conifer, 5%), forest (2%) and water surfaces (lakes and rivers, 3%) (see the supplement for the land cover map).

## 3 Data sets

### 3.1 Pléiades images

The satellites Pléiades-1A and 1B fly on the same near-polar sun-synchronous orbit at an altitude of 694 km with a 180° phase and descending node at 10:30 am. The CCD optical sensors acquire images in pushbroom mode by using 5x6000 pixel arrays and a maximum of 20 integration lines (TDI) for the panchromatic band ($480 - 830\,\mathrm{nm}$) (Poli et al., 2015). The system can achieve stereoscopic imaging with an additional quasi-vertical image (tri-stereoscopy), which is particularly suited for dense urban and mountainous areas. The tri-stereo mode can combine three stereo pairs to generate multiple DEMs, namely: front/nadir, nadir/back and front/back stereo pairs. The Pléiades' pixel depth at acquisition is 12 bits, and the panchromatic images have an initial resolution of 70 cm, but are oversampled at 50 cm by a post-processing algorithm that was implemented by the French Space Agency (CNES).

Two Pléiades triplets were acquired over the Bassiès catchment, which is the area of interest in this study (Tab. 1). The snow-free acquisition was programmed on 26 October 2014. Each snow-free image covered a surface area of approximately $117\,\mathrm{km}^2$, which was centred on the Bassiès catchment. The images were acquired with viewing angles of 11.9°, 0.7° and -10.9° in the along-track direction with respect to the nadir and -4.8°, -4.3° and -3.7° in the across-track direction. Consequently, the base to height (B/H) ratios were 0.22 (front/nadir pair), 0.23 (nadir/back pair) and 0.45 (front/back pair). The northern slopes were exposed to large shadows (approximately 10% of the catchment area) and exhibit poor image contrast because of the sun's position during autumn (sun elevation 34 °, azimuth 167°). No saturation or cloudiness were observed in the snow-free images.

The second triplet was acquired on 11 March 2015, when the snow accumulation was presumably close to its maximum peak. Each winter image covered a surface area of approximately $115\,\mathrm{km}^2$, which was centred on the Bassiès catchment. The images were acquired with viewing angles of 10.5°, -0.7° and -14° in the along-track direction with respect to the nadir




and with viewing angles of 0.4°, -2.7°, and -6.4° in the across-track direction. Consequently, the estimated B/H were 0.22 (front/nadir pair), 0.26 (nadir/back pair) and 0.48 (front/back pair). The images had a very low cloudiness (<2%). Saturated zones represented less than 3% of the images and wee located almost exclusively along the southern-exposed slopes. The northern slopes also exhibited abundant shadows (approximately 5% of the catchment area), but these shaded areas with low contrast were less extensive than those in the snow-free acquisitions (sun elevation 41°, azimuth 157°).

## 3.2 UAV images

Two winged-UAV photogrammetric surveys were performed over a central subset of the Bassiès catchment ($3.15\,\mathrm{km}^2$) to determine the snow depth by DEM differencing (Tab. 1). The UAV was a real-time kinematic (RTK) ©eBee that was equipped with a 12 MP camera. The flight altitude was maintained at approximately 150 m, which provided a mean ground sampling distance (GSD) from 0.10 to 0.40 m. The winter survey was performed on 10 March 2015. The snow-free survey was performed on 13 July 2015. Both the winter and snow-free acquisitions were achieved under very clear sky conditions. Onboard RTK corrections were performed at 20 Hz frequency. Georeferencing was performed during the winter through the use of a GPS-base, which was installed near the mountain refuge during the survey. Five georeferenced ground targets were placed in the valley bottom during the summer, and identified on drone images to improve the absolute positioning accuracy.

## 3.3 Snow probing

We collected approximately 501 hand-probed depth measurements on 10 March 2015, on the same day as the UAV survey, and one day before the Pléiades acquisition (Tab. 1). Because of the limited available time on the field, we attempted to cover an area that could represent of a large part of the catchment topography. The distance between each sample ranged from 10 to 30 m. We used two types of snow probes with length of 2.2 m and 3.2 m, respectively. The snow probing coordinates were recorded by using a differential GPS (DGPS) with a mean of 15 acquisitions (one per second) per probe location. We used the Trimble Geo XH 2008 (GPS) and Geo XH 6000 (GPS and Glonass). Post-treatment corrections were collected from a base that was 21 km away, specifically the French "Réseau Géodésique Permanent" network (RGP, base: "Mercus-Garrabet"). This process enabled us to achieve 0.1-m accuracy in the horizontal and vertical directions of the snow probing locations.

## 3.4 Land cover map

A 2008 land-cover map, which was updated by a field survey in July 2015, was generated through an object-based approach and expert-interpretation of aerial photographs (Sheeren et al., 2012; Houet et al., 2015) (see the supplement for the land cover map). The vegetation types were aggregated into seven classes to reflect the type of land cover that may influence the photogrammetric process: mineral surfaces (bare soil and rocks), water surfaces (rivers and lakes), peatland, low grass (rangeland, grassland, and subalpine meadows), shrubs, trees (conifer and deciduous), and unknown.



## 4 Methods

### 4.1 Production of DEMs, orthoimages and dDEMs from Pléiades images

A tri-stereoscopic acquisition was considered to i) limit the areas that were masked by the rugged topography of the studied catchment, ii) improve the correlation by providing different B/H ratios, and iii) obtain a nearly nadir image to improve the ortho-rectification process and accuracy of the absolute co-registration offset.

Snow-free and winter Pléiades DEMs were generated from the image triplets through the Ames Stereo Pipeline (ASP, version 2.4.8.), an open source automated stereogrammetry software by NASA (Broxton and Edwards, 2008; Moratto and Broxton, 2010; Willis et al., 2015) (Fig. 4). The ASP was primarily designed to create DEMs of ice and bare-rock surfaces. The ASP supports any Earth imagery that uses the RPC camera model format. Spatio-triangulation was based on the Rational Polynomial Coefficients (RPCs), which were provided as meta-data by Airbus Defense and Space (ADS), without block adjustment (i.e., no tie points). We parameterized the ASP to project the images into an epipolar geometry to reduce the search range before the correlation (Normalized Cross Correlation) and triangulation steps. We generated three point clouds from the three stereoscopic pairs from the *stereo* command. The DEMs were rasterized at 1-m, 2-m and 4-m cell sizes from the three raw point clouds through the *point2dem* command. The elevation values at a given grid point were obtained as a weighted average of the elevations of all points in the cloud within the search radius of the grid point, with the weights given by a Gaussian (see the supplement for the ASP's parameters) (NASA, 2015).

Four-meter snow-free and winter DEMs were horizontally co-registered by iteratively shifting the winter DEM with respect to the summer DEM (reference) by minimizing the standard deviation (SD) of the elevation difference distribution (Berthier et al., 2007). The final horizontal shifts were applied to the winter DEM were: $-5.2$ m in northing and $+2.8$ m in easting. This result is consistent with the expected localization precision that was provided by the RPCs from the Pléiades images. Without ground control points (GCPs), the horizontal location accuracy of the images was estimated at 8.5 m for a circular error at a confidence level of 90% (CE90) for Pléiades-1A and 4.5 m for Pléiades-1B (Lebegue et al., 2010; Gleyzes et al., 2013). The same shift was applied to the 2-m and 1-m winter DEMs.

Winter and snow-free nadir images were rectified at 1-m resolution for their respective DEMs, before co-registration. By picking 6 wide-spread corresponding points on the snow-free and winter images, the mean shifts were: $-5.2$ m in northing (SD=0.7 m) and $+3.2$ m in easting (SD=0.5 m), which are consistent with shifts from the DEM co-registration technique. The low SD values indicate that the horizontal shift was almost constant in the image. A classification of the image pixels into snow and snow-free classes based on intensity thresholds was performed on the winter ortho-image (Tab. 2).

dDEMs were produced at 1-m, 2-m and 4-m spatial resolution by subtracting the snow-free DEM from the winter DEM on a pixel by pixel basis:

$$\Delta Z_0 = Z_w - Z_s \tag{1}$$

where $Z_w$ is the pixel value in the winter DEM, and $Z_s$ is the pixel value is the snow-free DEM.



An absolute horizontal shift in the Pléiades DEMs was estimated from six wide-spread points that were identified on an aerial orthophoto from ©IGN("Institut National de l'Information Géographique et Forestière"), which presents an absolute accuracy of approximately 2 m. The shift between the snow-free Pléiades ortho-image and the IGN ortho-photo was: $+3$ m (SD=0.38 m) in northing and $-0.8$ m (SD=0.35 m) in easting. The dDEMs were then shifted based on this absolute horizontal offset, to be consistent with the DGPS and the georeferenced snow-probe measurements.

Then, we removed a constant vertical bias from $\Delta Z_0$ (Eq. 1) to obtain the final dDEMs:

$$\Delta Z = \Delta Z_0 - b \tag{2}$$

where $b$ is a constant vertical bias, which is determined from a unique, stable, and flat area of the photo that is easy to interpret. We chose to evaluate $b$ from a snow-free football field in the image that was 5 km from the mountain refuge (Fig. 1). The value of $b$ was assumed to be equal to the median of the dDEM distribution on the football field. After this bias correction, dDEM pixels with negative values were classified as "no data", which include 8 to 10 pixels that correspond to a snow probe measurement (Tab. 3). We classified the percentage of negative dDEM pixel values over the Bassiès catchment according to the presence of snow, and excluded shadow areas from steep rocks or cliffs.

Verifying whether a vertical bias that is measured over a small portion of a dDEM at low elevations (football field) can be used to correct an entire dDEM is very important. To test this assumption, we extracted 78 wide-spread values from the 2-m-Pléiades dDEM before bias correction (Eq. 1). We photo-interpreted these points on snow-free rock areas, roads or bare soil in the absolute geo-referenced winter ortho-image by avoiding the steepest slopes ($< 30°$), and by covering a large elevation range ($790 - 2510$ m). We did not use this information to remove the bias because we aimed to evaluate a simple workflow that could become operational.

## 4.2 Production of UAV DEMs and dDEMs from the UAV images

UAV DEMs were generated from the overlapping drone images (70% end lap, 70% side lap) by using the ©PIX4D software, which uses a structure-from-motion (SfM) algorithm (Westoby et al., 2012). Generated point clouds were rasterized at 0.1-m, 1-m and 2-m cell sizes for both the snow-free and winter DEMs. Subsequently, 0.1-m, 1-m and 2-m-dDEMs were obtained by differencing the corresponding snow-free DEM from the winter DEM. After an initial comparison with the sow probe measurements, a marked planar bias–oriented SW-NE was identified on the dDEMs. Comparing the winter UAV DEM values to the winter DGPS measurements (N=343) showed that the bias resulted from a bad stereo orientation, which led to some deformations in the winter DEM. To correct that bias, we extracted 353 wide-spread values from the 0.1-m-UAV dDEM at locations where the snow depth was supposed to be zero based on the winter ortho-image (emerging bare rock). We generated trend surfaces of order 1, 2 and 3 based on these values, and subtracted them from the 0.1 m-dDEM. The results improved significantly at each polynomial order, so we corrected the dDEM with the order 3 trend, which best fit the dDEM values from the emerging bare rocks (RMS before trend removal: 0.96 m, order 1 RMSE: 0.44 m, order 2 RMSE: 0.39 m, order 3 RMSE: 0.34 m). The results presented below are based on the de-trended dDEM values at each pixel resolution. An extra point that



was located on the heliport of the mountain refuge was used to correct a constant bias after the trend removal (0.1 m: +0.33 m, 1 m: +0.43, 2 m: +0.41 m).

## 4.3 Pléiades and UAV dDEMs assessments and comparison

We compared the Pléiades dDEM at 1-m, 2-m and 4-m resolutions and the UAV dDEM at 0.1-m, 1-m and 2-m resolution to the snow probe measurements. We calculated the values of the residual vector $R_{\Delta Z}$ as follows:

$$R_{\Delta Z} = \Delta Z_i - HS \qquad (3)$$

$\Delta Z_i$ is the subset of the dDEM values, where $\Delta Z \geq 0$ after bias correction (Eq. 2), which were sampled by snow probing. $HS$ are the snow-probe measurements. We considered that the measurements from the snow probes had a random error of $\sigma_{probe} = 0.15\,\mathrm{m}$, but did not introduce a systematic error term.

The metrics that were used to describe the quality of the dDEMs were the percentage of "no-data" values after the stereo processing and the statistics of $R_{\Delta Z}$: (i) the mean and the median, which were used to evaluate the vertical accuracy of the dDEMs, and (ii) the standard deviation (SD) and the normalized median absolute deviation (NMAD), which were used to characterize its vertical precision. The NMAD is a metric for the dispersion of data that is not as sensitive to outliers as the SD (Höhle and Höhle, 2009):

$$\mathrm{NMAD} = 1.4826 \cdot \mathrm{median}(|(R_{\Delta Z} - m_{R_{\Delta Z}}|) \qquad (4)$$

where $m_{R_{\Delta Z}}$ is the median of the errors.

We also assessed how the $\Delta Z$ and $HS$ values correlate through a rank correlation method. We used the Spearman correlation factor, called $cor_s$, which is neither sensitive to the presence of outliers nor the existence of nonlinear correlations (Chueca et al., 2007; Borradaile, 2013).

The snow depth was greater than the snow probe length for 50 occurrences. These cases where the operator did not reach the ground were excluded from these statistics, and were only exploited as binary information to assess the dDEMs (see the supplementary materials).

We snapped and subtracted the 2-m-UAV dDEM from the 2-m-Pléiades dDEM. We visually compared both dDEM maps and the dDEM differences. We performed a SW-NE transect (1.6 km long) and compared the dDEM values along that transect.

## 4.4 Residual analysis on the Pléiades data

### 4.4.1 Photogrammetric processing

We calculated the density of the summer and winter raw point clouds that were generated during the correlation process based on the front nadir/stereo pair (Fig. 4). The Pléiades panchromatic images had a pixel size of 0.5 m, so a mean density of 4 points per square meter would indicate a correlation success at the minimum achievable interval. Areas with lower density values require a higher search range in the interpolation of the raster cell value from the point cloud.





### 4.4.2 DEM contributions

To identify whether the final errors were due to the snow-free DEM or the winter DEM, we computed two distinct residuals terms for the 2-m-Pléiades dDEM as follows:

$$R_{Z_w} = Z_w - Z_{w,\mathrm{DGPS}} \tag{5}$$

$$R_{Z_s} = Z_s - (Z_{w,\mathrm{DGPS}} - HS) \tag{6}$$

The error on the DGPS measurements was $\sigma_{DGPS} = 0.1\,\mathrm{m}$ for all the points. The second term of the equation 6 ($Z_{w,\mathrm{DGPS}} - HS$) provides snow-free reference elevation values at the snow-probe locations. This term has an error from to both uncertainty in the snow probe measurement and the DGPS error. Hence, two error terms exist because of the DGPS and snow-probe measurements ($\sigma_{DGPS+probe} = 0.18\,\mathrm{m}$, see the supplement for more details on the error calculation).

### 4.4.3 Snow height, topography and land cover influences

Various factors limit the acquisition of snow probe measurements, such as exposure to avalanches, human mobility on this challenging terrain, and the available time on the field. The snow depths that were obtained with the snow probes ranged from 0 to 3.2 m(Fig. 3). We assessed the influence of HS on the residuals between the dDEM and HS (equation 1). The snow heights from the snow probes do not represent the entire topographic variability of the catchment (Fig. 3). Here, we summarize the different ranges of the main topographical variables that are associated with the snow probe data:

- The sampling-elevation range is 1645 – 2000 m a.s.l. However, 70% of the snow-probe values are between 1645 and 1700 m a.s.l. The elevation range of the catchment is 1156 – 2676 m a.s.l. (median elevation of 1930 m). Therefore, we did not assess the residuals' distribution (equation 1) according to the elevation.

- The slope, which was derived from the 2-m-snow free Pléiades DEM, was associated with the snow-probe measurements and ranged continuously from almost 0°(flat areas) to 25°. A dozen snow-probe values were recorded in steeper zones but were not considered as statistically representative. The median slope of the catchment was 26°, and a variety of slope values are present in the catchment, from flat area to cliff.

- The different aspect classes were well sampled during the snow probe survey.

- The snow depth sampling range according to the curvature was quite limited because of the difficulty in performing snow probing in marked convex or concave areas. Therefore, we did not assess the residuals distribution (equation 1) according to the curvature.

The distribution of the residuals between the dDEM and HS values was analysed according to the different land cover classes. The land cover classes in the snow probe data were: minerals (12%), water surfaces (4%), low grass (32%), shrubs (33%) and peatland (19%).The peatland class is overrepresented and mineral surfaces are underrepresented in the probe dataset with respect to the Bassiès catchment area.





## 4.5 Contribution of the tri-stereoscopy

To our knowledge, the added-value of tri-stereoscopy relative to bi-stereoscopy has not been clearly established for an open alpine terrain. To provide a preliminary assessment of this contribution, we generated two seasonal DEMs from two individualized stereo pairs in both snow-free and winter cases. The first Pléiades pair consists of backwards and almost nadir images, and the second pair consists of forward and almost nadir images. Consequently, we generated a dDEM map for each stereo pair. We compared these dDEMs to the snow-probe measurements.

## 5 Results

### 5.1 Pléiades dDEMs assessments

The snow-free DEM and winter DEM are shown in Fig. 5. The small-scale topographic features are well captured by the high-spatial resolution of the DEMs. The winter DEM is characterized by a smoother texture. The distribution of the dDEM values (inset in Fig. 6) has the typical gamma or log-normal distribution shape that is reported in the literature (e.g. Winstral and Marks, 2014). Considering the whole Bassiès catchment, the mean of $\Delta Z$ is $2.15\,\mathrm{m}$ and its standard deviation is $1.72\,\mathrm{m}$.

The Pléiades 2-m-dDEM is composed of 1.7% of "no data" entries in the Bassiès catchment (2.4% and 1.2% for the 1-m and 4-m-dDEMs, respectively) (Tab. 2). These "no-data" entries originate from data gaps in the raw point coulds, which are produced by the ASP before rasterization. Considering the Bassiès catchment area, 25% of the pixels of the 2-m-dDEM exhibit negative values (23% and 22% for the 1-m and 4-m-dDEMs, respectively). The percentage of negative 2-m-dDEM pixel values on the snow-covered area is 17 % (14.7% and 14.5% for the 1-m and 4-m-dDEMs, respectively). This fraction is less important if we do not consider the snow pixels in the shaded areas (direct shadow from the surrounding cliffs): 11.3 % for the Pléiades 2-m-dDEM (9.4% and 9.8% for the 1-m and 4-m-dDEMs, respectively). The Pléiades 2-m-dDEM pixels with values above $15\,\mathrm{m}$ represent a very limited fraction, which is negligible on snow (less than 0.1 %). These values should most probably be interpreted as inconsistent and classified as no data.

We calculated a constant vertical bias $b$ from a snow-free football field (see section 4.1). The value of $b$ for each dDEM resolution is $b_{1\,\mathrm{m}} = -0.46\,\mathrm{m}$, $b_{2\,\mathrm{m}} = -0.48\,\mathrm{m}$, and $b_{4\,\mathrm{m}} = -0.44\,\mathrm{m}$. The bias value distribution of the football field has a mean value that is close to the median (1 m: $-0.43\,\mathrm{m}$, 2 m: $-0.45\,\mathrm{m}$, and 4 m: $-0.42\,\mathrm{m}$) and a low standard deviation (SD, 1 m: $-0.25\,\mathrm{m}$, 2 m: $-0.2\,\mathrm{m}$, 4 m: $-0.15\,\mathrm{m}$). The bias assessment which was performed over the entire Pléiades dDEM ($110\,\mathrm{km}^2$) and was based on 78 wide-spread values (see section 4.1) indicates a median of $-0.70\,\mathrm{m}$, a mean of $-0.74\,\mathrm{m}$ and an SD of $0.26\,\mathrm{m}$. The low SD value and the median difference confirm the possibility to remove a constant bias from a unique area, with small random and systematic errors:

$$\mathrm{median(football\ field)} - \mathrm{median(entire\ dDEM)} = -0.22\,\mathrm{m}.$$

The comparison with the snow probe data indicates that the Pléiades dDEMs are consistent with the snow depth measurements (Tab. 3). The median values of the residuals distribution $R_{\Delta Z - HS}$ are relatively low (between $-0.12\,\mathrm{m}$ and $-0.16\,\mathrm{m}$) and





close to the mean of the distribution at each pixel resolution ($\pm 0.05$ m between the median and mean). A slight influence from the pixel size is present (Tab. 3). For our validation dataset, the 2-m-Pléiades dDEM exhibits slightly better precision and accuracy. For this dDEM, the SD is $0.58$ m and the NMAD is $0.45$ m. The $\Delta Z$ and $HS$ datasets are significantly correlated at each pixel resolution ($cor_s$ ranges between 0.67 and 0.72). The linear regression between the dDEM values and the snow probe measurements is close to the 1:1 line ($\Delta Z_{2m} = 0.90 \cdot HS$) (Fig. 7).

Figure 8 shows the spatial distribution of the snow depth measurements and the residuals of the 2-m-dDEM. No obvious pattern is present in the residuals, although the absolute residuals are higher in the southern part of the surveyed area, where the slopes are the steepest (see Sect. 5.4.3).

Overall, the snow probe dataset exhibits a low systematic error and is spatially homogeneously distributed.

## 5.2 UAV dDEMs assessments

The standard deviation and the NMAD indicate a decimetric random error, at each pixel resolution ($SD_{2m}$ =0.62 m, $NMAD_{2m}$=0.35 m). The $\Delta Z$ and $HS$ values are significantly correlated (mean $cor_s$ 0.79). The median value of the residual distribution $R_{\Delta Z - HS}$ is slightly negative and ranges from -0.07 to -0.15 m according to the pixel size.

Figure 8 shows the spatial distribution of the snow depth measurements and the residuals of the 2m-UAV dDEM. From this map, no obvious pattern is present in the residuals.

## 5.3 Comparison of the Pléiades and UAV dDEMs

The Spearman correlation factor $cor_s$ between the Pléiades and UAV dDEM values is 0.62 and significant at 95% confidence (N=527.10³). The dataset were not co-registered. The comparison between the 2-m-Pléiades and the 2-m-UAV dDEMs is characterized by a residual distribution with a median of $-0.14$ m (mean $-0.06$ m), an SD of $1.47$ m, and an NMAD of $0.78$ m.

The 2-m-Pléiades and UAV dDEM maps exhibit very similar patterns (Fig 9). Similar snow features are identifiable in both dDEM maps, such as a marked over-accumulation of snow along a topographic ridge that stretches from the refuge to the lake, snow traps for wind-blown snow and snow cornices. These features are also observable in the terrestrial photography (Fig 2). A transect over a common area that is covered by both the 2-m-Pléiades and 2-m-UAV dDEMs highlights the consistency in both $\Delta Z$ variations. Over this transect, the SD of the residuals between the Pléiades and UAV dDEMs is $0.53$ m and the median is close to $0$ m.

## 5.4 Residual analysis on the Pléiades data

### 5.4.1 Photogrammetric processes

The density values of the raw point clouds (pts. m²) from the correlation process based on the front nadir/stereo pair are close to the maximum achievable value (4 pts. m²) in both the winter and snow-free DEMs at the snow-probe locations (see the



supplement for the density map). Therefore, the dDEM assessment should not be influenced by the interpolation process that creates the raster DEMs at the first order.

### 5.4.2 Pléiades DEMs assessment

We decompose the respective contributions from the snow-free and winter DEMs to the dDEM residuals (equations 5 and 6, Fig. 10). The medians of $R_{Z_w}$ and $R_{Z_s}$ distributions are $-0.91\,$m and $-0.25\,$m, respectively, leading to a difference of:

$$\mathrm{median}(R_{Z_w}) - \mathrm{median}(R_{Z_s}) = -0.66\,\mathrm{m}.$$

This value is consistent with the median of $-0.64\,$m for the $R_{\Delta Z_0}$ distribution that was identified with the HS probe measurements before the bias correction (the bias that was identified on the football field was -0.48 m). The $R_{Z_w}$ and $R_{Z_s}$ values in Fig. 10 are corrected from the bias by removing the median. The SDs of $R_{Z_w}$ and $R_{Z_s}$ are 0.32 m and 0.66 m, respectively. These estimations are consistent with the standard deviation of the 2-m-residual distribution $R_{\Delta Z}$ (0.58 m).

### 5.4.3 Influences of topography and land-cover

The correlations between the residuals distribution and the snow depth or the terrain slope are weak but significant (0.3 and 0.26). The deviation of the residuals distribution $R_{\Delta Z}$ increases slightly with the slope. However, the number of snow-probe measurements varies by interval and thus limits the interpretation of the statistics (Tab. 4).

The snow probe measurements associated to the low grass and peatland classes present the lower deviation in the residuals distribution (SD, 0.49 and 0.51 m). The most important dispersions are associated to the mineral and the shrub classes (SD, 0.79 and 0.63 m).

## 6 Discussion

### 6.1 Production of DEMs and dDEMs from Pléiades images

The method that was proposed here is based on VHR satellite stereo imagery. The agility of the Pléiades satellites provides a wide range of B/H ratios, including small values, which are necessary for alpine topography. We programmed a B/H of 0.2 between two consecutive stereo pairs to improve the correlation success rate and limit the shading effect of topography. The snow-free and winter front/back pairs (B/H=0.4) created less dense photogrammetric clouds. Thus, the number of "no-data" pixels would have increased in the final DEM for a bi-stereo acquisition that was based on a B/H of 0.4 instead of 0.2.

The stereo-orientation from the RPC ancillary data was sufficient to adjust the relative orientation of the images prior to their projections in the epipolar geometry. The affine epipolar transform of both the left and right images is based on automated tie-point measurements, the effect of which is equivalent to rotating the original cameras which took the pictures (NASA, 2015). The command *bundle adjust* could probably improve the relative stereo orientation. We did not intentionally use ground control points to avoid the need for a field survey in the workflow. We did not remove outliers from the 3D triangulated point cloud, which could be done by parameterizing the ASP ("near and far universe-radius parameters", see the supplement).



The images that show which pixels were matched by the stereo correlator, which are called "good pixel maps" in the ASP, highlight a significant correlation in both the snow-free and winter DEMs. For steep slopes and/or a limited density of raw photogrammetric clouds, the map-projection of the images through the *mapproject tool* on a coarse DEM before the *Stereo pre-processing* stage of ASP could improve the correlation success. Another option could be the direct calculation of the distance between the snow-free and winter raw point clouds instead of a raster representation (Westoby et al., 2015; Passalacqua et al., 2015).

The statistics, which were calculated separately for both DEMs, highlight the better performance in the elevation determination of the snow-covered images compared to the snow free images (Fig. 10). This observation could be due to the difficulty in treating micro-topography with the native GSD of Pléiades (0.7 m at nadir). Snow-covered areas offer a smoother surface compared to vegetated or stony snow-free surfaces. The results on bare rock may be directly connected to the slope influence because most of this type of surface is located on steep slopes (Tab. 4). In both the snow-free and winter acquisitions, the shadow areas were the most challenging for the correlation process and appeared as very noisy surfaces with more "no-data" entries because of the correlation failures and outliers, such as negative dDEM values after vertical bias removal. The resolution of 2 m presents the most favourable statistics according to our validation dataset and potentially highlights a good compromise between the horizontal accuracy and the smoothing of the snow height.

Snow areas under shadows from high cliffs constitute a large erroneous fraction of negative dDEM pixel values (Tab. 2). Together with emerging steep rock, these areas should be treated as no-data entries with a sufficient buffer to limit the uncertainties on the mean HS retrieval.

No snow fall occurred in the Bassiès catchment during the 20 hours between the field survey and the Pléiades acquisition. Fresh snow probably may have complicated the correlation stage and increased the number of saturated pixels. During the triangulation stage, we did not exploit the multi-view stereo possibility of the ASP (only available since version 2.5.0), which limited our correlation to successive pair matching. Berthier et al. (2014) showed for the Mont-Blanc area that a simple combination of the different DEMs derived from the three images of a tri-stereo can reduce the percentage of data voids and slightly improve the precision of the merged DEM. In our case, we did not notice an improvement in the dDEM precision through the comparison with the snow probe measurements (SD=0.69 m for the tri-stereo 4-m-dDEM, SD=0.64 and 0.61 m for the bi-stereo). The accuracy was slightly better for the tri-stereo dDEM (median=-0.12 m for tri-stereo, median=-0.54 and +0.13 m for bi-stereo). The medians were of opposite signs for the front/nadir and nadir/back stereo pairs, which may explain the median values for the tri-stereo. The density maps from the point clouds exhibited similar patterns, because the correlation failed for both stereo pairs in the shadow areas.

## 6.2 Comparison to the snow-probe measurements

The validation dataset was strongly limited by the measurement protocol. To cover the largest extent in a limited time, we did not apply an optimal sampling strategy to assess the entire snow depth variability at a plot scale, typically 10 m x 10 m (López-Moreno et al., 2011; Bühler et al., 2015). The dDEM pixel values were therefore assessed by a unique snow depth measurement, which could explain the modest correlation between the dDEM values and the snow probe measurements (mean





$cor_s$ = 0.7 for Pléiades). The snow probes were too short to measure the highest snow depth, and we only provided binary information in these cases (see the supplement). We did not survey the highest crest where drifted snow accumulates, which led to the highest snow accumulations. Even with longer snow probes, sampling the snow depth in these areas would not have been safe . Increases in slope have a clear influence on the magnitude of the dispersion of the residuals between the dDEMs and

the snow probe measurements. However, the snow probe dataset was not sufficiently representative to determine the influence of the slope.

### 6.3    Comparison to the UAV dDEM

A bias was identified in the winter UAV-DEM. We could remove this bias in the final UAV-dDEM thanks to the snow-free bare rock areas, which provided a valuable opportunity to generate widespread vertical offsets. However, this strategy for bias

correction has obvious limits, and identifying and correcting the sources of these errors would have been better. The RTK signal was repeatedly lost during the survey, which negatively affected the photographs' orientation. The acquisition mode of UAV photographs is largely "non-convergent", which could also result in marked deformation (Westoby et al., 2015). Winged UAVs are less stable than UAVs with rotors, and their potential for snow mapping in high-alpine catchments must be further assessed. We noted large mismatches between the Pléiades and UAV dDEM maps for steep slopes, which could be due to incorrect flight

plans or, lens calibration, co-registration errors (James and Robson, 2014). However, despite these discrepancies, we consider that the UAV dDEM map was a valuable independent source to evaluate the Pléiades snow depth map because the comparison revealed similar snow depth patterns.

### 6.3.1    Limitations and perspectives

The digital photogrammetric determination of snow depth in mountainous areas has been a longstanding issue (Cline, 1993,

1994; Ledwith and Lundén, 2001). Until recently, terrestrial and aerial photographs and optical satellites images have been used almost exclusively to determine the spatial distribution of snow cover area (SCA). Identifying conjugate and ground control points and contrast and lighting issues were the main factors that have impeded the production of DEMs of snow-covered areas.

Recent works have highlighted the potential of airborne-derived techniques to produce centimetric and decimetric vertical accuracy and precision in DEM generation over snow-covered areas and in dDEM generation from snow-free and winter

DEM differencing. Pléiades-derived snow heights do not have the same accuracy and precision compared to this state-of-the-art of digital aerial photography. Lee and Jones (2008) have created a DEM over a snow-covered area of a mountainous terrain in Australia from a high-spatial-resolution camera (GSD up to 0.05 m) and an enhanced radiometric resolution (12 bits) on a GPS/Inertial motion unit (IMU) airplane system. An assessment by 183 GPS measurements revealed a mean error of +0.14 m with an SD of 0.08 m. Bühler et al. (2015) employed an opto-electronic line scanner (ADS 80) that was mounted

on an aeroplane to map the snow depth at 2-m-resolution (GSD 0.25 m) in the Swiss Alps. A comparison between the ADS and different individual HS measurements revealed both RMSE and NMAD of approximately 0.3 m, which is equivalent to 1 GSD of the input images. Over the polar snow of Alaska, Nolan et al. (2015) generated dDEMs over rather flat areas from a consumer-grade camera that was coupled to a dual-frequency GPS on a manned aircraft without the use of an IMU.





The comparison of the dDEMs to 6000 snow-probe measurements highlighted an SD of the residuals of $0.1\,\text{m}$ (GSD 0.06 to $0.2\,\text{m}$). A DSLR camera that was mounted on a UAV platform was used over a small mountainous terrain ($0.07\,\text{km}^2$) with thick vegetation cover, Jagt et al. (2015) to map the snow depth at a very high spatial resolution (GSD $6.10^{-3}\,\text{m}$). A comparison to 20 snow-probe measurements highlighted an RMSE of $0.096\,\text{m}$ using GCPs, and $0.184\,\text{m}$ without ($0.084\,\text{m}$ with one point of

5 co-registration). These techniques that are based on airborne platforms remain suitable if clouds are present above the flight altitude. However, these approaches present serious constraints absent from satellite acquisition: the need for a pilot, a ground operator, or the use of a specific sensor and an ad hoc installation. In remote areas such as high mountain catchment, these requirements could seriously compromise the acquisition process.

Pléiades, along with GeoEye-1, WorldView-1, WorldView-2 and QuickBird, belongs to class 6 satellites (0.40 - $0.75\,\text{m}$

GSD). The main limitation of the images that are derived from these satellites could be the surveying of large areas because of the relatively limited swath ($20\,\text{km}$ for Pléiades). The maximum length of Pléiades stereoscopic coverage from the same orbit with a B/H of 0.2 is $80\,\text{km}$ for a stereo acquisition and $25\,\text{km}$ for a tri-stereo acquisition ($195\,\text{km}$ and $80\,\text{km}$, respectively, for a B/H of 0.4) (Gleyzes et al., 2012). Considering a B/H of 0.2, areas of up to $1600\,\text{km}^2$ may be imaged repeatedly in any part of the world that is covered by the Pléiades satellite constellation. Pléiades images do not exhibit the best spatial resolution of

this class. However, its main advantage is its pixel depth at acquisition of 12 bits, while other VHR sensors have a pixel depth at acquisition of 11 bits. With 4096 shades of grey by pixel instead of 2048, subtle nuances, especially at the beginning or end of the spectrum, can be distinguished. As for all optical sensors, the main drawback of the Pléiades constellation is the need for clear-sky conditions to obtain suitable cloud-free images. Snow-free images can be acquired over a large temporal window, and repeating these acquisitions each time a dDEM must be processed is unnecessary. Winter images are more constrained

because the key moment to evaluate the snow cover height is the vicinity of the accumulation peak, which may span several weeks. However, the daily revisit interval of the Pléiades satellite constellation increases the possibility of obtaining cloud-free and valuable images. Winter data sets can also be acquired at the end of various winters for inter-annual comparisons of snow-depth.

The method that was proposed here does not provide any information on the snow thickness under trees. The ALS remains

the only technique to extract high-resolution HS information in forested terrain. In the study area, this point is not critical because most of the catchment is open terrain. In general, most of the snow in the Pyrenees accumulates above the tree line near $1600\,\text{m}$ a.s.l. (Gascoin et al., 2015).

Despite the above mentioned limitations and given the results of this first study, we believe that satellite photogrammetry is a promising alternative to recently developed techniques that are based on Lidar or aerial digital photogrammetry to retrieve

snow depth. This conclusion is especially true in areas where field or airborne campaigns are not feasible or too expensive and where the snow accumulation is significant (above $2\,\text{m}$). In glaciology, DEMs that are generated from optical stereos are often considered to be inaccurate in accumulation areas (Schiefer et al., 2007; Racoviteanu et al., 2010). However, Pléiades DEMs that are acquired at the beginning and end of accumulation seasons could be used to evaluate the seasonal components of the glacier mass balance (Berthier et al., 2014). In hydrology applications, there remains a substantial uncertainty on the

final snow volume at the watershed scale that need to be better assessed. In our study site, the mean dDEM value in the Bassiès





catchment area ($14.45\,\mathrm{km}^2$) was 2.15 m. The corresponding CV value was 0.80 (the ratio of the SD to the mean snow depth). This CV agrees with the classification that was proposed by Liston (2004) since it falls in the category 9 "mid-latitude, treeless mountain (e.g. Rocky Mountains, alpine)". In terms of accumulation, the 2011-2012 winter was very comparable to the 2014-2015 winter in the Bassiès catchment. According to a Meteo-France meteorological reanalysis, the precipitation was $1130\,\mathrm{mm}$

over the hydrological year 2011-2012 and $1150\,\mathrm{mm}$ in 2014-2015. Szczypta et al. (2015) used a distributed snowpack model to simulate the snowpack and its temporal evolution on a regular grid over the Bassiès catchment at a spatial resolution of 25 m during the 2011–2012 snow season. At the accumulation peak, the mean monthly snow depth that was simulated over the entire catchment in April was 2.2 m. Although both mean values cannot be readily compared, the order of magnitude appears to be consistent with the mean dDEM value that was found for the 2014-2015 winter and was based on Pléiades data.

## 7   Conclusions

We generated a DEM difference map that was based on winter and snow-free tri-stereoscopic Pléiades satellite images. The comparison of this Pléiades dDEM map to 451 snow probe measurements, which were collected simultaneously, shows that the snow height can be retrieved from space with decimetric systematic and random errors and a metric horizontal resolution at the scale of a small mountain watershed ($14.5\,\mathrm{km}^2$). The distribution of the residuals between the 2-m-Pléiades dDEM values

and the corresponding snow probe measurements present a median of $-0.16$ m and an SD of 0.58 m. An independent dDEM map was generated through a winged UAV photogrammetric survey on the same date based on a similar workflow. Despite some outliers, the UAV dDEM map was also successfully validated by the snow probe measurements (median of the residuals is $-0.11$ m, SD is 0.62 m). The snow cover features that were obtained by Pléiades DEM differencing were consistent with those that were derived from the UAV acquisition. The correlation between the snow heights from both techniques is statistically

significant, even if some discrepancies were present on the steepest slopes.

Further studies should focus on influences of the snow height, the topography and the land cover on the accuracy of Pléiades-derived snow heights based on Lidar-derived snow height maps. Our validation dataset limited the analysis to gentle slopes or relatively flat areas and snow heights up to 3.2 m. The shadows that are projected onto slopes create a lack of radiometric contrast in both snow-free and winter images and constitute an inherent limitation to optical sensors. Other limitations include

obstructions by the forest canopy and cloud cover.

These results are promising because they open the possibility to retrieve the snow height at a metric horizontal resolution in remote mountainous areas that are difficult to access. Indeed, our workflow does not require any field measurements because collecting ground control points is unnecessary. The size of the study area could vary from several square kilometres to several hundreds of square kilometres.

*Acknowledgements.*  The Pléiades images acquisition was supported by the CNES through the ISIS Pléiades programme (Projet CRYOPYR–4500107601). This work was supported by the Région Midi-Pyrénées and the University of Toulouse through the CRYOPYR project. We thank the OHM Vicdessos Human-Environnemental Observatory for logistic and financial support. We also warmly thank Audrey Chone,



Yoann Malbeteau, Yoann Moreau, and Vincent Rivalland who helped us to collect the snow-probe measurements. The authors deeply acknowledge Michael J. Willis for his help in ASP parameterization to produce the DEMs. We also thank Joaquin Maria Munoz Cobo Belart for its useful comments that improved the manuscript.



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





**Figure 1.** Top: localization of study site relative to Europe, and to the Pyrenees mountain. Bottom: Bassies catchment $(14.5\,\mathrm{km}^2)$ on a Top 25 topographic map from ©IGN ("Institut national de l'information geographique et forestière") map.





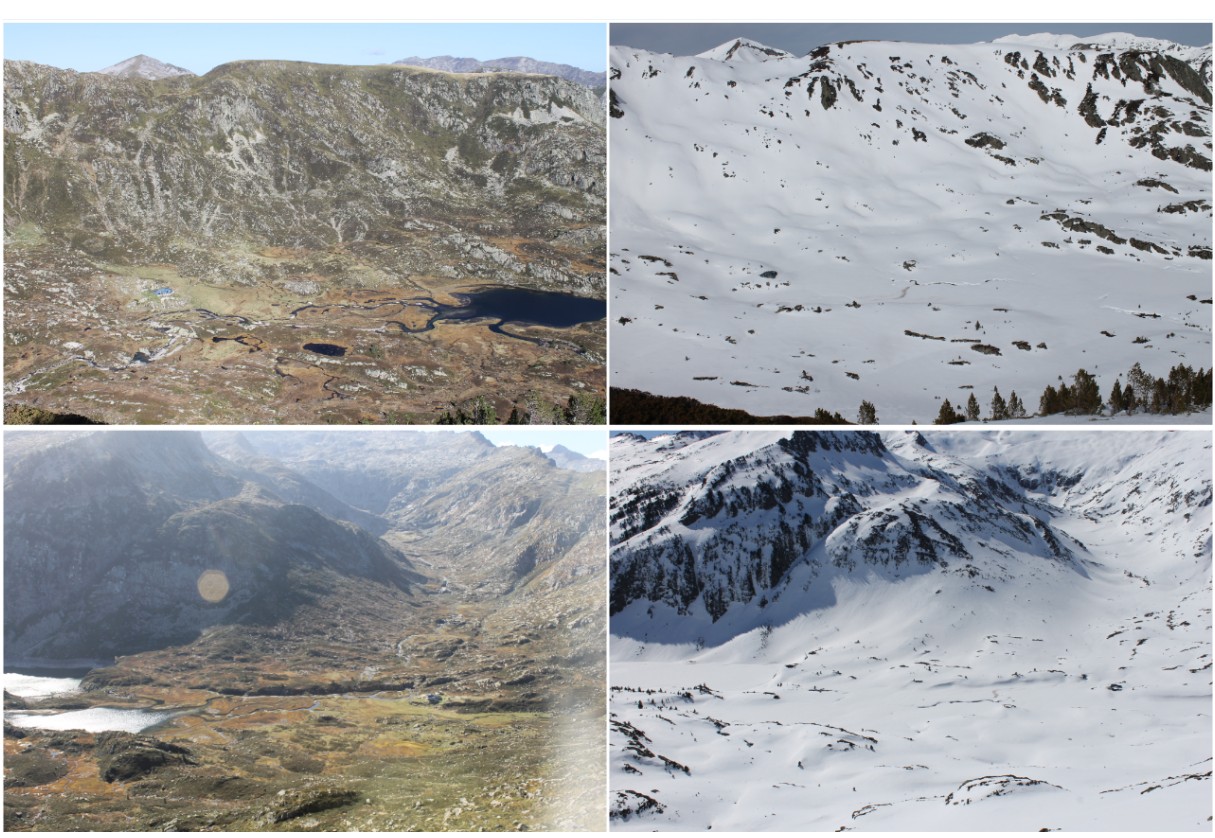

**Figure 2.** Comparison of oblique pictures taken by automatic camera (see Fig. 1 for localization). On the left, the pictures were taken on 26 October 2014. On the right, the pictures are from 10 March 2015. These photographs were taken at the same time of day as the Pléiades images taken on 11 March 2015 (11:00 LT).





**Figure 3.** Distribution of the snow probe sampling, according to the snow depth, the elevation, the slope, the aspect, the curvature and the land cover classes. On the whole snow probe sampling (N=501), the snow probe did not reach the ground for 50 occurrences.





| Acquisition | | Metrics |
|---|---|---|
| snow-free and winter images triplets localized by RPC model | | GSD, CE 90 (m) |

| Correlation | | |
|---|---|---|
| Correlation on the epipolar images in AMES Stereo Pipeline | | Correlation rate |
| snow free and winter XYZ clouds | | cloud density (pts.m$^{-2}$) |
| Rasterization | | |
| snow free 1,2,4 m-DEM | winter 1,2,4 m-DEM | Pixel size |

| Registration | | |
|---|---|---|
| Horizontal DEM coregistration | | X,Y shifts (m) |
| Ortho-image absolute registration | | ($\sigma_x$ ; $\sigma_y$) |
| Vertical bias removal | | Z shift (m) |
| Absolute georefenced DEMs and DEM difference | | |

| Validation | | |
|---|---|---|
| Comparison with DGPS and snow probes | | Statistics on residuals (m): |
| | | $R_{\text{snow free DEM}}$ $R_{\text{winter DEM}}$ $R_{\text{dDEM}}$ |
| Residuals assessment by slope classes and land cover type | | |
| Comparison with UAS dDEM | | $R_{\Delta\text{dDEMS}}$ |

**Figure 4.** Successive phases in Pléiades triplets processing, and associated metrics of assessment.



**Figure 5.** Hillshade of snow free 2 m-Pléiades (top-left) and 1 m-UAV DEM (bottom-left) DEMs, and winter 2 m-Pléiades (top-right) and 1 m-UAV (bottom-right) DEMs.





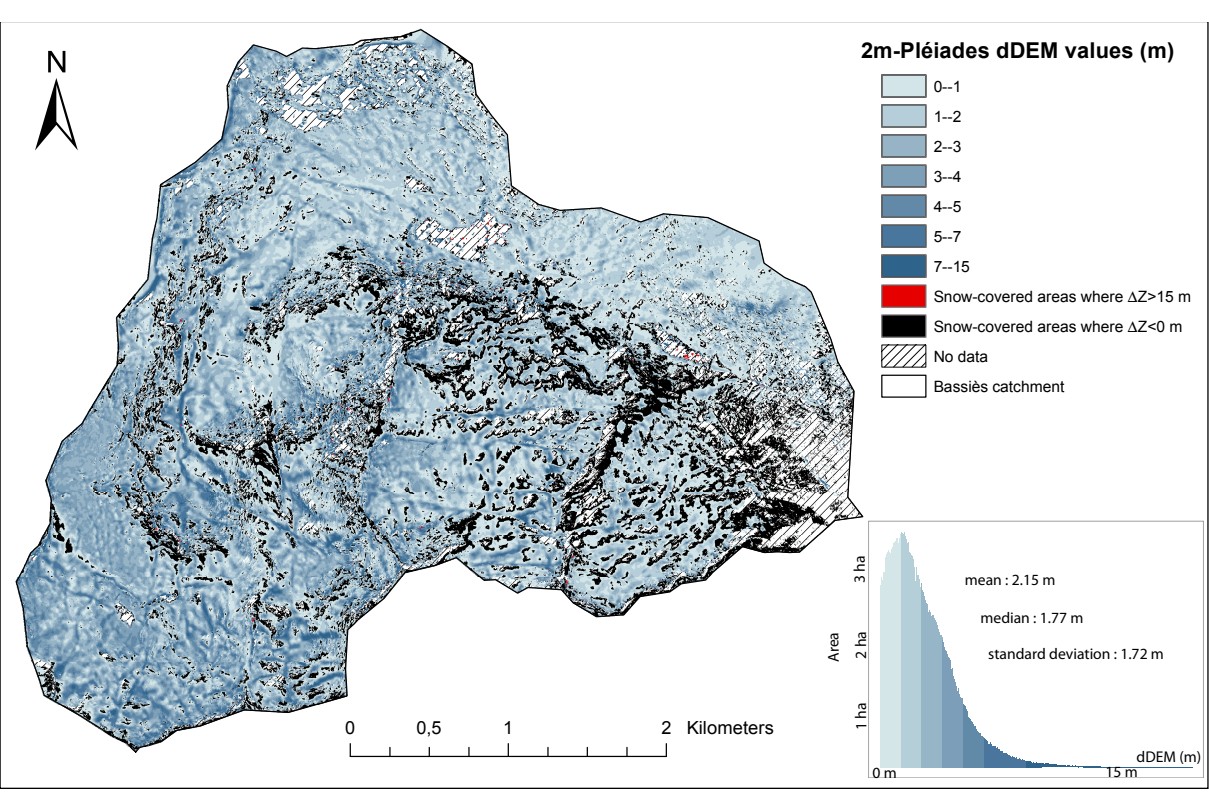

**Figure 6.** 2 m-Pléiades dDEM map interpreted as snow height ( m) in Bassies catchment.




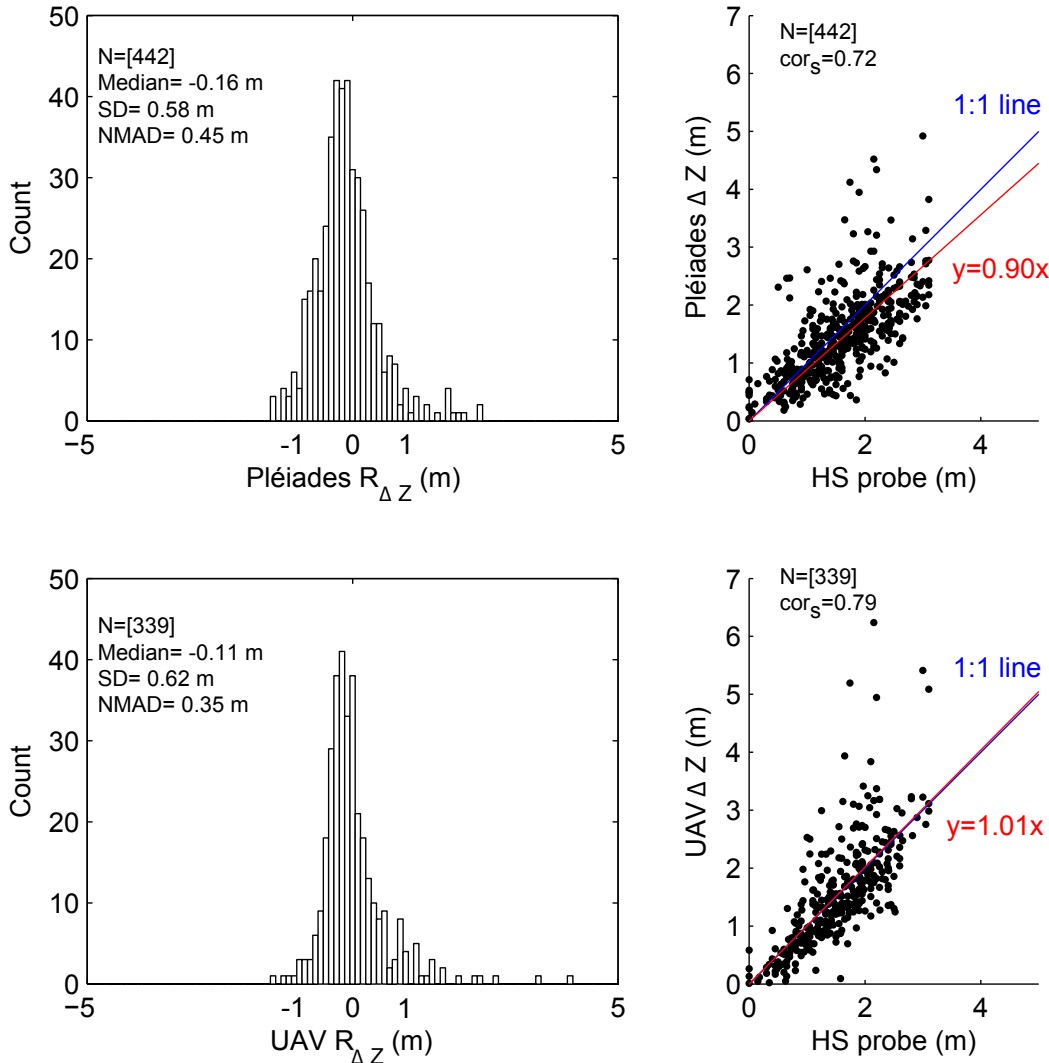

**Figure 7.** Top left: histogram of the distribution of the residuals between the 2 m-Pléiades dDEM and the snow-probe measurements. Bottom left: histogram of the distribution of the residuals between the 2 m-UAV dDEM and the snow-probe measurements. Top right: scatter plot between the 2 m-Pléiades dDEM and the snow probe measurements. Bottom left: scatter plot between the 2 m-UAV dDEM and the snow probe measurements. Blue line is the 1:1 line. Red line is the least-square best fit of a linear function with a zero intercept $y = ax$.





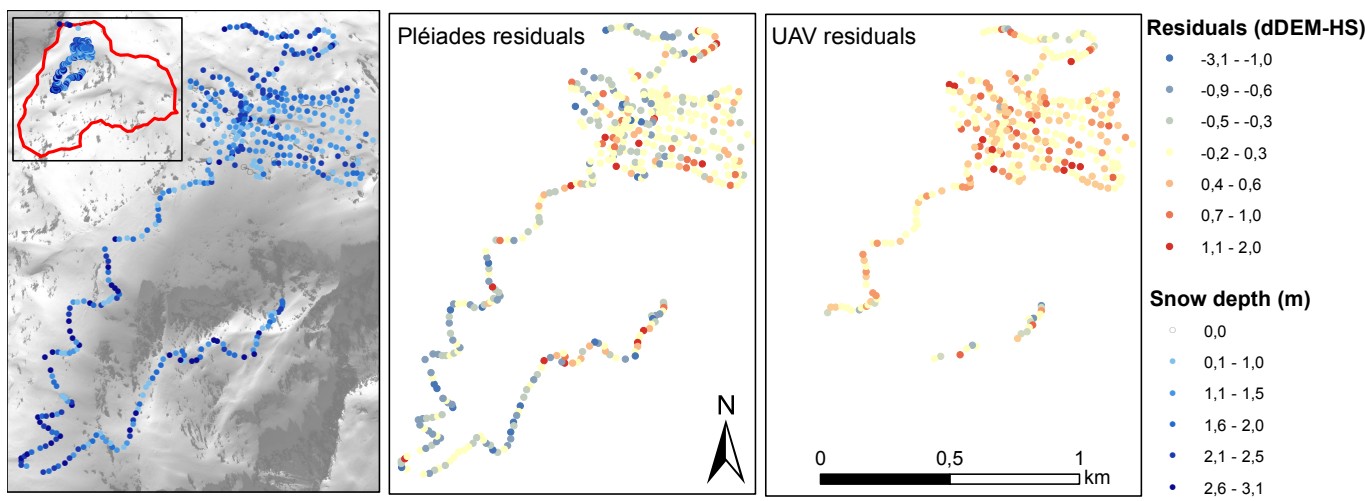

**Figure 8.** Left: map of snow depth sampling over the Pléiades winter ortho-image. Middle: map of 2 m-Pléiades residuals $R_{\Delta Z}$. Right: map of 2 m-UAV residuals $R_{\Delta Z}$.





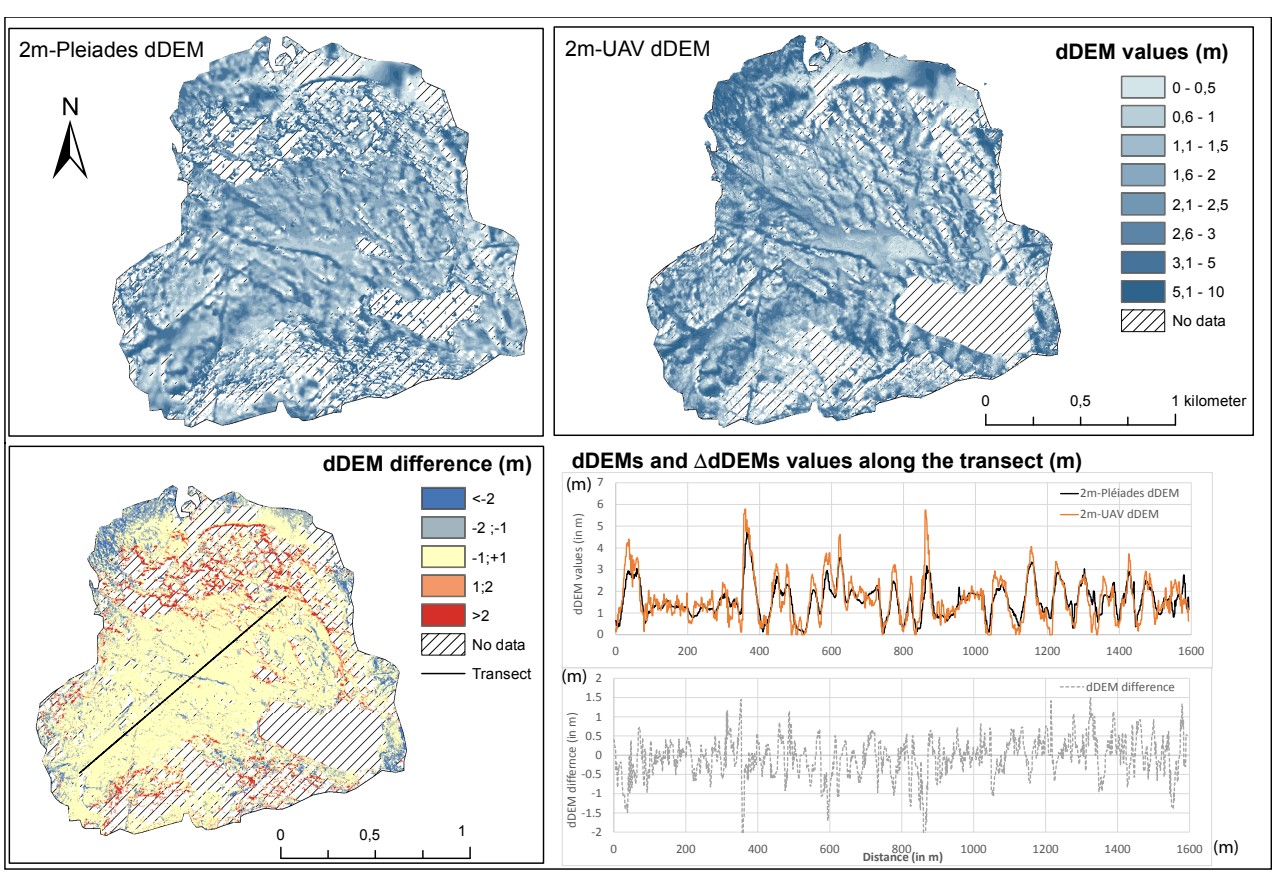

**Figure 9.** Pléiades and UAV dDEM, and dDEM differences. dDEMs and ΔdDEMs (dDEMs differencing) values along the transect.





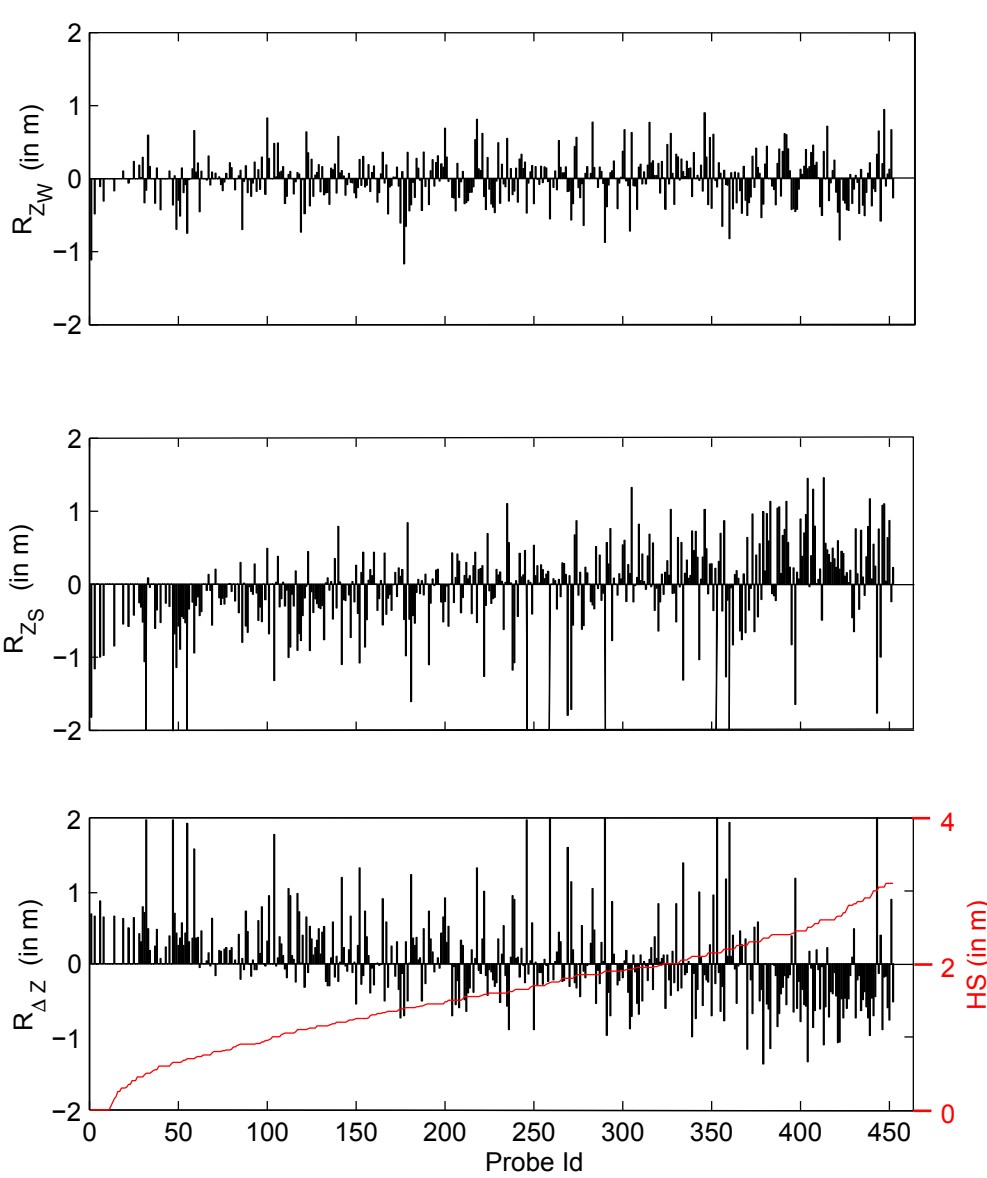

**Figure 10.** Residual error terms, after the systematic error removal, for the 2 m-Pléiades winter DEMs, the snow-free 2 m-Pléiades DEM, and the 2 m-Pléiades dDEM according to the probe Id ranked in the ascending HS order (see equation section 4).





**Table 1.** Data sources and description. ADS means "Airbus Defence and Space". GEODE and CESBIO are both laboratories of the Toulouse University (France). ©GeoFalco is a French start-up specialized in UAV data acquisition and processing.

| Data sources | Acquisition date | Institution (acquired by) | Ground Sampling Distance (m) | Photogrammetric information | Products (resolution) |
|---|---|---|---|---|---|
| 1B-Pléiades triplet | 26 Oct. 2014 | ADS | 0.70 – 0.73 m | B/H=0.22; 0.23; 0.45 | snow-free DEM (1 m; 2 m; 4 m) |
| Snow probe measurements | 10 March 2015 | GEODE CESBIO | 10 – 30 m | - | validation dataset |
| UAV photographs | 10 March 2015 | GeoFalco | 0.10 – 0.40 m | 70% end-lap, 70% side lap | winter DEM (0.1 m; 2 m) |
| 1A-Pléiades triplet | 11 March 2015 | CNES | 0.70 – 0.73 m | B/H=0.22; 0.26; 0.48 | winter DEM (1 m; 2 m; 4 m) |
| UAV photographs | 13 Jul. 2015 | GeoFalco | 0.10 – 0.40 m | 70% end-lap, 70% side lap | snow-free DEM (0.1 m; 2 m) |



**Table 2.** Percentage of potential outliers and no data in the dDEM values, considering the catchment area, the snow-covered area of the catchment, and the snow-covered area of the catchment located out of the shadows due to the high cliffs (called here below "sunny snow").

| Data source | Pixel size | No data | Percentage of | | | | |
| --- | --- | --- | --- | --- | --- | --- | --- |
| | | | $\Delta Z < 0\,\mathrm{m}$ | | | $\Delta Z > 15\,\mathrm{m}$ | |
| | | | in the catchment | on snow | on sunny snow | in the catchment | on snow |
| | 1 m | 2.4 % | 22.4 % | 14.7 % | 9.4 % | 0.14 % | 0.09 % |
| Pléiades tri-stereo | 2 m | 1.7 % | 24.5 % | 17 % | 11.3 % | 0.15 % | 0.1 % |
| | 4 m | 1.2 % | 22 % | 14.5 % | 9.8 % | 0.17 % | 0.1 % |



**Table 3.** Statistics relative to the comparison between the Pléiades and the UAV dDEMs to the snow probe measurements, according to the pixel resolution. Significant correlations (p values <0.05) are marked with asterisks.

| Data source | dDEM pixel size | Number of snow-probe sampling | Median (m) | Standard deviation (m) | NMAD (m) | Spearman correlation $cor_s(\Delta Z, HS)$ |
|---|---|---|---|---|---|---|
| Pléiades tri-stereo | 1 m | 443 | -0.15 | 0.62 | 0.47 | 0.71* |
| | 2 m | 442 | -0.16 | 0.58 | 0.45 | 0.72* |
| | 4 m | 441 | -0.12 | 0.69 | 0.51 | 0.67* |
| Pléiades front/nadir stereo pair | 4 m | 411 | -0.54 | 0.64 | 0.53 | 0.62* |
| Pléiades nadir/back stereo pair | 4 m | 450 | 0.13 | 0.61 | 0.47 | 0.73* |
| UAV photographs | 0.1 m | 343 | -0.07 | 0.63 | 0.38 | 0.8* |
| | 1 m | 336 | -0.15 | 0.62 | 0.36 | 0.79* |
| | 2 m | 339 | -0.11 | 0.62 | 0.35 | 0.79* |





**Table 4.** Statistics relative to the comparison between the 2m-Pléiades dDEM (tri-stereo) and the snow probe measurements, according to the snow depth, slope and aspect, and the land cover classes.

| Variable | Interval bins | Number of snow-probe sampling | Median (m) | Standard deviation (m) | NMAD (m) | Spearman correlation $cor_s(|R_{\Delta Z}|, HS)$ |
|---|---|---|---|---|---|---|
| Snow depth | [0 ; 0.5 m] | 25 | 0.24 | 0.22 | 0.31 | 0.3* |
| | ]0.5 m; 1 m] | 65 | -0.01 | 0.46 | 0.33 | |
| | ]1 m; 1.5 m] | 94 | -0.07 | 0.44 | 0.39 | |
| | ]1.5 m; 2 m] | 114 | -0.24 | 0.60 | 0.34 | |
| | ]2 m;2.5 m] | 72 | -0.32 | 0.68 | 0.54 | |
| | ]2.5 m; 3.2 m] | 46 | -0.63 | 0.56 | 0.39 | |
| Slope | ]0°; 5°] | 150 | -0.10 | 0.42 | 0.32 | 0.26* |
| | ]5°; 10°] | 117 | -0.19 | 0.53 | 0.41 | |
| | ]10°; 15°] | 81 | -0.30 | 0.53 | 0.59 | |
| | ]15°; 20°] | 63 | -0.30 | 0.79 | 0.7 | |
| | > 20° | 31 | -0.18 | 0.93 | 0.75 | |
| Aspect | North | 159 | -0.20 | 0.6 | 0.43 | - |
| | East | 113 | -0.15 | 0.63 | 0.48 | |
| | South | 134 | -0.16 | 0.55 | 0.46 | |
| | West | 43 | -0.12 | 0.55 | 0.39 | |
| Land cover | All classes | 442 | -0.16 | 0.58 | 0.47 | 0.72 |
| | Mineral | 56 | -0.2 | 0.79 | 0.60 | 0.74 |
| | Water | 21 | -0.32 | 0.55 | 0.50 | 0.67 |
| | Low grass | 140 | -0.16 | 0.49 | 0.35 | 0.74 |
| | Shrub | 140 | -0.15 | 0.63 | 0.51 | 0.68 |
| | Peatland | 84 | -0.15 | 0.51 | 0.42 | 0.69 |