# Peer review of "Mapping snow depth in open alpine terrain from stereo satellite imagery"

_The Cryosphere, 2016_

## Referee Comment (RC1) · Y. Bühler (Referee) · 23 Feb 2016

The paper entitled "Mapping snow depth in open alpine terrain from stereo satellite imagery" by R. Marti et al. investigates the potential of very high spatial resolution (VHR) optical satellite imagery for snow depth (HS) mapping in an alpine catchment. This investigation is to my knowledge the first attempt using such data for this purpose and is therefore a significant contribution for many different potential applications.

The achieved precisions of the snow depth values compared to manual probe and UAV measurements are approximately 0.5 m. This is slightly better than the 0.7 m spatial resolution of the input imagery and therefore in line with other investigations applying digital photogrammetry from airplanes (Bühler et al. 2015, Nolan et al. 2015) and UAVs (Bühler et al. 2016, Harder et al. 2016, Vander Jagt 2015). In my opinion this

contribution should be published after taking into account the following comments:

1. During the process of generation the snow depth maps and its evaluation, systematic offsets between the summer and winter DSMs as well as the reference datasets are eliminated. These x,y and z-offsets are crucial for the final product as they influence the error by 100% or more. The calculation of these offsets and their elimination is described in the text. However, it is very hard to follow and to understand. I propose that you generate an overview in a table or a figure where you list the different offsets, their amount and how they were eliminated. What information is necessary to eliminate them?

2. I do not understand the comparably low precision (SD 0.6 m) of the UAV reference data set even though the correlation and the NMAD are better than for the Pleiades HS values. Recent studies report accuracies of approximately 0.1 m (Bühler et al. 2016, Harder et al. 2016, Vander Jagt 2015). Was the problem saturation of the imagery? Even though the RTK signal was lost the relative accuracies within the DSM should be much better. Please explain this issue in more detail and relate your results to the recent studies mentioned here.

3. Compared to the manual reference data you achieve an underestimation of the HS values of approximately 0.15 m (median). Is there an explanation for this? Could it come from uprising summer vegetation such as bushes that are pressed down to the bottom by the snowpack as reported in Bühler et al. (2016)?

4. You state that the major benefit of the Pléiades sensor is its 12-bit radiometric resolution compared to 11 bits of other comparable satellite sensors. I doubt this statement. I do not think that there is a significant benefit of 12-bit data compared to 11-bit data (while there should be one compared to 8-bit sensors!). Are really all 4096 digital numbers used? In my experience also 11-bit data never uses the whole dynamic range. Could you show some histograms of the input imagery? I would guess that you get very similar results using 11 bit data. With 11/12 bit data you should get good results

Interactive
comment

in shadowed areas but you have to mask them out. Can you explain why you do so?

5. There should be a table listing all available and planned satellite sensors that could potentially be applied for HS mapping including their temporal, spectral, radiometric and spatial resolution.

6. In my opinion there should be a discussion of potential important applications. For what applications the identified precision of 0.5 m is sufficient? What are the applications where you need better precision for example generated from UAV or laser scanning data?

Technical corrections:

P1L1: there is passive microwave; you describe this later in the paper.

P1L2: optical stereo satellites

P1L12: please give the calculated precision vs. the UAV data here

P1L14: I think it is very dangerous to propose the application of remote sensing data without any field data! You need at least some reference measurements to be sure your values are OK. I really suggest deleting this statement!

P1L23: From my experience it is more wind, snow avalanches and terrain features that generate the high spatial variability of alpine snow depth distribution.

P2L27: The main advantage of near-nadir looking instruments against TLS is that you have no holes caused by terrain features such as ridges or bumps and that you can cover the entire area spatially continuous.

P3L29: UAS?

P5L6: It would be nice if some more details on the UAS campaign could be given here. How many images were acquired? What camera did you use? . . ..

P5L16: approximately 501?

P6L13: Why did you choose these spatial resolutions? Please justify.

P6L15: Gaussian distribution?

P6L17: What are the drawbacks of this method? The winter and summer surface is not similar due to the snow cover. Why is this approach still working? Or did you only use snow free areas to do the SD calculations? Why did you use the 4 m DEMs? This is not clear to me.

P6L27: Can you give some more details on the classification? What happens if snow is in shadow areas?

P7L11: I think it is a bit dangerous to sent negative snow depth to no data as you might change the statistics significantly. There might also be many false values, which are slightly positive. These values might differ out more or less. Can you discuss this point and give some indications?

P7L26: What do you mean by bad stereo orientation? I do also not completely understand you approach with the trend surfaces. These points need a better description.

P8L9: How do you get to the error value of 0.15 m for probe measurements? Please justify. I would assume it is much less, something around 0.05 m.

P8L24: (Fig. 9) P10L15: Where and why do you get these data gaps in the point clouds?

P10L18: 527.103 what entity is this? Also pts.m2 throughout the document.

P10L19: SD and NAMD are pretty bad compared to the other results. Can you explain why? The NMAD Satellite/Probe is 0.45 m and the NMAD UAV/Probe is 0.35 m. Are they both shifted in the opposite direction? Or how can you get to a NMAD of 0.78 m?

P12L17: Why?

P14L15: Discuss your results in the context of the results published by Harder et

al.(2016), Bühler et al. (2016) and Vander Jagt et al. (2015).

P16L1: Please explain CV

P16L18: Please mention the result of the Satellite / UAS HS comparison

P16L28: This statement is dangerous! You need at least had probe measurements in the file to get an idea about the achieved accuracies.

P16L29: What is the outreach of these results compared to HS measurements with LiDAR, airplanes and UAV? What are potential applications?

P27 Fig. 5: Please indicate the outline of the UAV extent in the Pleiades extent.

P28 Fig. 6: The chosen bins are too wide. If you change to a continuous color scale ranging over more than one color, you can make much more details visible. Please adapt the color scale.

P31 Fig. 9: Please change the color scale as in Fig. 6. Also the error bins are too wide in my opinion. You only see the very large errors of more than one meter like this. Could you set the profile from one end to the other, like this you do not display the big errors in the northeast because you stop just before that.

P32 Fig. 10: How do you get errors compared to the probe measurements for the summer DSM? I do not really understand the figure caption, please clarify.

P36 Tab4: Why is there only one cos value for all snow depth classes? What does the star mean? The same for the slope classes and the aspect classes.

References:

Bühler, Y., Adams, M. S., Bösch, R., and Stoffel, A.: Mapping snow depth in alpine terrain with unmanned aerial systems (UAS): potential and limitations, The Cryosphere Discuss., 2016, 1-36, 2016.

Bühler, Y., Marty, M., Egli, L., Veitinger, J., Jonas, T., Thee, P., and Ginzler, C.:

Snow depth mapping in high-alpine catchments using digital photogrammetry, The Cryosphere, 9, 229-243, 2015.

Harder, P., Schirmer, M., Pomeroy, J., and Helgason, W.: Accuracy of snow depth estimation in mountain and prairie environments by an unmanned aerial vehicle, The Cryosphere Discuss., 2016, 1-22, 2016.

Vander Jagt, B., Lucieer, A., Wallace, L., Turner, D., and Durand, M.: Snow Depth Retrieval with UAS Using Photogrammetric Techniques, Geosciences, 5, 264-285, 2015.

---

## Referee Comment (RC2) · E. THIBERT (Referee) · 4 Mar 2016

Review on :

**Mapping snow depth in open alpine terrain from stereo satellite imagery**
by R. Marti and co-authors,
The Cryosphere Discuss., doi:10.5194/tc-2016-11

E. Thibert,
Grenoble - 04/03/2016.

*General comment*
R. Marti and co-authors employ advanced methods of tri-stereo high resolution satellite imagery to retrieve DEMs and snow-cover thickness by DEMs differentiation in a mountainous open terrain. In a rigorous comparison with ground-based manual snow-probing, and UAV-derived DEMs differentiation, the authors find generally good agreement in these comparisons, the discrepancy been explainable by the natural scattering of the data, but also by some residual biases that remained unexplained. These favourable results should confer significant advances and improvements in mapping the snow mantle thickness over large open areas.
The paper is almost clear, well organized, and properly focuses the scope of the journal.

The paper would be much more valuable were it also to provide a much clear presentation of the bias corrections in both xy and z-vertical directions for the DEMSs derived from Pléiades and UAV flights. The resulting improvements in the residuals before/after adjustments are not clearly displayed (but dispersed along the text). A summary would be welcome with the numerical values of adjustments as an additional table or a supplementary column in Figure 4.

Regarding the results of the comparisons between measurement methods, some questions remain:
- Is the remaining bias (-0.12 — -0.16 m) between the Pléiades estimation of snow thickness relative to manual probing significantly different from 0 from your error analysis?
- Same question between Pleiades and UAV DEMs (-0.14m)?
A physical explanation should be attempted if you conclude theses biases to be significant.

Regarding specifically the UAV image acquisition and process from automatic correlation structure-from motion, much more needs to be said about the image orientation, whether the camera model was estimated from a self-calibration in the process, or fixed from standard values or an independent prior calibration.

A minor point that arises throughout the paper is the indistinct use terms of error in systematic and random meanings, which can sometimes be confusing for the reader. To avoid confusion, I would as much as possible use systematic error, bias, and discrepancy to quantify the incorrectness, and random error and scattering to denote inaccuracy. Moreover, the random error should be defined with respect to the standard deviation (one or 2 times the standard deviation for example). This will help you to decide about the differences between the results to be notified as significant or not. Random errors are combined assuming they are uncorrelated (page 5 of the supplement). This should be stated in the core of the text.

An additional figure would be helpful, setting z-vertical direction and the different almost-horizontal surfaces $Z_s$ $Z_w$ from Pléiades and UAV and GPS, to define the notations of the variables, their differences, and better understand how standard deviations combine in the equations of the supplement.

Here follow some detailed questions, comments, suggestions, and indications of minor typos in the paper.

*Substantive comments*

P1-L2. Specify ground resolution in meters for consistency with the rest of the paper (all lengths in meter elsewhere)

P1-L8. To me synonyms are accuracy and precision. Is it meant accuracy (random error) and correctness (bias or systematic error)?

P1-L21. I would set at first that snow cover is important in those areas for live and ecosystems, and second for anthropogenic needs.

P1-L16. The time for the seasonal snow thickness peak is very dependent on the elevation, varying from December at less than 1000 m a.s.l., to March in the 2000—2700 m range and April-May above 3000 m.

P1-L23. The natural spatial variability of the snow cover thickness is preliminary due to the variability in precipitations, and post-deposition processes as wind drift, avalanches, snow densification.

P4-L20. use units in metres for consistency with the rest of the paper

P4-L20. As far as I know, specify that this oversampling is carried out before image delivery

P4-L23. specify the local time.

P5-L11—14. It should be clarified between winter and autumn surveys, what set up is used among RTK, ground control points.

P5-L23. The z-vertical correctness and accuracy is generally less than that of the xy plane, especially a vertical bias is unavoidable without a geoid model or an independent altimetry adjustment on levelling points.

P5-L28. Make it explicit whether your refer to influences in radiometry or ground surface roughness.

P4-L8. Figure 4 introduced before Figure 3 (in page 9 L14)?

P4-L8—11. Define RCP acronym just after "…Earth imagery that uses the RPC…" instead of the next sentence.

P4-L13. It should be clarified that each of the three resolution DEMs is retrieved from a rasterisation of the 3 point cloud corpus (otherwise it may be understood that you derive one raster DEM per point cloud).

P4-L10. It should be clarified by a few words that RCP sets the image-to-ground geometry.

P4-L15. I would rewrite as: "…grid point, with the Gaussian curve as weighting function…"

P4-L17. Which is the RMS residual in z after co-registration?

P4-L23. Is it meant that the same xy shift is applied to the higher resolution DEMs without proceeding to a new minimization? Why? And why the 4 m-resolution as the resolution to proceed for all DEMs?

P5-L7—14. Please give more details about the UAV-on-board camera.
It is presumably a non-metric camera. Is it a fixed focal length lens camera? What is the focal length? How on board RTK corrections to tag image centre coordinates and ground control points are used jointly for orientation, and maybe the camera self calibration?
I suspect 5 ground control points insufficient for a calibration.

P7-L3. Are you really sure that the IGN ortho has a more correct xy referencing than your DGPS?

P7-L6. Does that bias mean than the co-registration was not optimal? How compares this value with the co-registration residual?

P7-L8. Which photo? Does this refer to the satellite winter and autumn images?

P7-L11. I am not sure it is correct to remove negative snow depths as these values may not be significantly different from zero considering random errors in both you snow probing and dDEM calculations. They may well be acceptable in terms of confidence interval.
Removing some values is nevertheless conceivable considering they might be abnormal (irregular) if they discard negatively from zero at 2 times sigma (or more), according the way you define "aberrant" values.

P7-L15—19. What is the result for the vertical bias calculated from these 78 points. Even if unused later in the paper, how does it compare to the bias calculated from the football field?

P7-L21—23. A bit more needs to be said about the image orientation (calibration?) process from the UAV acquisition.
Particularly, calibration for non-metric cameras is known to be critical and can generate significant orientation error when processed through automatic correlation Structure-from-Motion'based software as used here.
Which camera model is used for the orientation process? Did you used a simultaneous self-calibration or a prior calibration? If calibrated, which camera parameters are estimated (decentration, radial distortion and associated polynomial coefficients—how many? , focal length)?
Can you give the orientation residuals? This will help you to discuss about the discrepancy between Pléiades and UAV results to be significant or not.

P8-L9. It should be explained how $\sigma_{probe}$ is estimated as it is surprisingly high.

P8-L16. Instead of error, I would write "…is the median of the dDEM/probe discrepancies.".

P9-L11. Section 4.4.3. From the section 3.4, I expected here an analysis of the ground surface roughness effect, both on manual snow probing and in $Z_s$ uncertainty from the 2 DEMs.

P10-L31. How can you interpret physically this remaining systematic bias for the Pléiades snow thickness estimation? Is it significantly different from zero from your error analysis?

P11-L5. Why did you force the intercept to be zero and did not fit to Y=aX+b in search of a systematic difference?

P11-L11. I would expect a much lower value for the residual from the UAV DEM.
Can you comment this in relation to the orientation residual of the images and the overall quality of the DEM geometry?

P11-L12. The subscript for the residual R is inconsistent with notations from equations 3, 5, 6 ?

P14-L8—17. I would also question the bias you identified here in relation to the quality of the geometry of the UAV DEM to originate from the orientation of your images.
More needs to be said about cameras, camera models, and the statistics of orientation results. Instability is frequently associated to inaccurate or residual correlated camera model parameters after least-square adjustment (none unicity of solutions) which can result in poor quality of the geometry of the 3-D model (such as doming or bowl effects) after image orientation.

P14-L27. Is "dynamic" much more appropriated than "resolution" to denote the 12-bits depth?

P14-L28. To make a better distinction between accuracy/correctness concepts, use bias instead of "error" when you refer to a systematic error. Or systematically qualify errors as random or systematic to avoid confusion.

P15-L18. Inflect somewhat writing "…for clear-sky/limited cloud cover conditions…"

P15-L34. "In hydrology **and water resource** applications, there remains…"

P16-L8. This result is not trivial. The snow cover thickness you can estimate is not significantly different from that of the snowpack model, considering the overall uncertainty in both estimations.

P16-L26—29. I would mitigate/inflect your conclusions here mentioning that you nevertheless need an altimetry control/adjustment on a snow-free flat surface —as your z-vertical bias corrections demonstrate— that you have to infer on each satellite imagery. But this control surface can be located kilometres apart and at lower elevations.

*Supplement*

It is not clear to me why $\sigma_{probe}$ appears in equations 3, 4 and 5. I would only expect this term in the uncertainty associated to the comparison of dDEMs with HS.

*Figures*
P23-Figure 1 — caption. "Pyr**én**ées mounta**ins**. Bottom: Bassi**è**s…"
Identically, in the legends of top right and main maps : "Bassi**è**s"

P24-Figure 2 — caption. "Comparison of **terrestrial** oblique pictures taken by automatic **cameras**…"
The local time is mentioned here but not in the text.

P30-Figure 8 —left map. Add a title "Snow depth" at bottom right corner as you did for the 2 other maps at top left.

P32-Figure 10 — caption. Add plot colours "…and the 2m-Pléiades dDEM (black bars) according to the probe Id ranked in the ascending HS (red line) order (see equation section 4)."

P32-Figure 10 — labels. Point of notation: define the units for the residual errors as (m), and not (in m). Same for HS in the right-hand axis.

*Stylistic comments*
P1-L1.  At present…

P1-L19.  Snow Coverd Area…. Snow Height

P1-L20.  Snow Water Equivalent (SWE)

P4-L1. Bassi**è**s and Pyr**én**ées

P4-L7 "  6.6°C and the mean annual precipitation is 1640mm

P4-L9 "…and 25% by vegetation-free rock and bare soils."

P4-L30. erase "which was"

P4-L30. "…115km$^2$, and centred on the Bassiès catchments, as achieved for snow-free images."

P4-L32. "…-&4°  along track direction…"

P5-L1. "…-6.4°  across track direction."

P5-L9 "…mean Ground Sampling Distance (GSD)…".

P5-L13. "…installed in a nearby mountain refuge…"

P5-L16. "We collected up to 501 hand-probed…"

P5-L19. "…10 March 2015, at the time of the UAV survey…"

P5-L19. "…snow probes with length**s** of 2.2m and 3.2m,…"

P6-L3. "…to i) limit the areas **potentially** masked by the rugged topography…"

P4-L24. "…1-m resolution **from** their respective DEMs…"

P7-L24. "…s**n**ow probe"

P8-L1. "…the flat dropping zone of the mountain hut…"

P8-L15. Here and in lines 5 and 18 on page 11, don't know the correct sign to use for multiplication between dot and cross signs following the journal style? my preference is cross…

P9-L14. space to add between 3.2 m and (Fig.3).

P12-L27.  "tie-point measurements, whose effect is equivalent…"

P14-L4. space to erase after "safe"

P14-L28. In upper case letters "Motion Unit"

P15-L3. Unclear sentence to correct, it seams that "by" is missing before Jagt et al. (2015)?

P36-Table 3 — caption. Remind the reader what the acronym NMAD denotes.

P36-Table 4 — caption. It would be helpful to remind the reader that **\*** denotes significant correlations as mentioned in Table 3 caption.

P36-Table 4. What does reversed brackets denote in the interval bin column?

---

## Author Response (AR1)

Reviewer #1 : General comments

**Y. Bühler (Referee)**
buehler@slf.ch

The paper entitled "Mapping snow depth in open alpine terrain from stereo satellite imagery" by R. Marti et al. investigates the potential of very high spatial resolution (VHR) optical satellite imagery for snow depth (HS) mapping in an alpine catchment. This investigation is to my knowledge the first attempt using such data for this purpose and is therefore a significant contribution for many different potential applications. The achieved precisions of the snow depth values compared to manual probe and UAV measurements are approximately 0.5 m. This is slightly better than the 0.7 m spatial resolution of the input imagery and therefore in line with other investigations applying digital photogrammetry from airplanes (Bühler et al. 2015, Nolan et al. 2015) and UAVs (Bühler et al. 2016, Harder et al. 2016, Vander Jagt 2015). In my opinion this contribution should be published after taking into account the following comments:

We would like to thank Y. Bühler for his constructive comments and suggestions, and for the time he spent on our manuscript. We agreed with most of its comments as detailed below in the point-by-point response.

*(1) comments from Referees, (2) author's response, (3) author's changes in manuscript.*

1. During the process of generation the snow depth maps and its evaluation, systematic offsets between the summer and winter DSMs as well as the reference datasets are eliminated. These x,y and z-offsets are crucial for the final product as they influence the error by 100% or more. The calculation of these offsets and their elimination is described in the text. However, it is very hard to follow and to understand. I propose that you generate an overview in a table or a figure where you list the different offsets, their amount and how they were eliminated. What information is necessary to eliminate them ?

As suggested by the reviewer, we added a new table which provides an overview of the differents offsets identified and the adjustments performed accordingly. In the last column, we made a distinction between the values of adjustments performed during the workflow to produce the final product and the values of verification of these adjustments:

**Table 3.** Summary of the differents co-registrations and the bias corrections performed to produde the Pléiades and the UAV DEMs and dDEMS maps. SD means Standard Deviation.

| Input data | Reference data | Type of coregistration | Values of adjustments | Comments |
|---|---|---|---|---|
| 4 m-Pléiades winter DEM | 4 m-Pléiades summer DEM | xy relative coregistration | -5.2 m North +2.8 m East | Workflow data Same shifts applied to the 1 m and 2 m-Pléiades winter DEMs |
| 1 m-Pléiades winter ortho-image | 1 m-Pléiades summer ortho-image | xy relative coregistration | -5.2 m North +3.2 m East | Verification data |
| 1-2-4 m-Pléiades dDEMs | dDEM-snow free footbal field | z relative coregistration | $b_{1\,m} = -0.46\,m$ (SD=0.25 m) $b_{2\,m} = -0.48\,m$ (SD=0.20 m) $b_{4\,m} = -0.44\,m$ (SD=0.15 m) | Workflow data |
| 2 m-Pléiades dDEMs | 78 wide-spread points over snow-free areas | z relative coregistration | Median $b = -0.70\,m$ Mean $b = -0.74\,m$ SD $b = 0.26\,m$ | Verification data |
| 1 m-Pléiades summer ortho-image | 6 wide-spread points on the 0.50 m-IGN ortho-image | xy absolute coregistration | +3 m North (SD=0.38 m) -0.8 m East (SD=0.35 m) | Workflow data Same shifts applied to the dDEMS |
| 0.1 m-UAV-dDEM | 353 wide-spread points over snow-free areas | $\Delta$Z-correction based on a trend surface of order 3 | RMSE: 0.34 m | Post-treatment correction. Same correction applied on the 1 m and 2 m-UAV dDEMs |

The proposed workflow does not require any external data to generate the snow depth map since the co-registration of the snow/no-snow DEMs is based on the Pléiades data only: (i) horizontal translation based on the minimization of the dDEM standard deviation (ii) vertical shift to remove the elevation bias found on a snow-free surface (Fig. 4).

However, for this study we had to perform an additional geometric correction to allow the comparison of the dDEMs with our validation datasets (snow probes and UAV surveys). Here we needed an external reference dataset (IGN aerial ortho-image). Note that the absolute registration of the dDEM is not required if the snow map calculation method is applied in another site where such high-quality reference data are not available (although validation data are always most welcome!).

2. I do not understand the comparably low precision (SD 0.6 m) of the UAV reference data set even though the correlation and the NMAD are better than for the Pleiades HS values. Recent studies report accuracies of approximately 0.1 m (Bühler et al. 2016, Harder et al. 2016, Vander Jagt 2015). Was the problem saturation of the imagery?

Even though the RTK signal was lost the relative accuracies within the DSM should be much better. Please explain this issue in more detail and relate your results to the recent studies mentioned here.

The reviewer is right that the accuracy of the HS values retrieved by the UAV approach in our study is somewhat disappointing, especially given the results of the most recent studies with similar instruments. A lower NMAD value (NMAD=0.35 m) than  the SD value (SD=0.62 m) indicates that our UAV dDEM is affected by outliers.

In our case, we do not have a clear explanation on this issue, but we can only speculate that it is a combination of (i) the loss of the RTK signal (ii) the saturation of some areas (iii) the high values of the UAV camera view angle in some parts of the study area.

The main goal of our study is to evaluate if satellite data can play a role in snow depth mapping. To investigate this question, the  451 snow probe measurements is the foremost validation dataset We did not use the UAV dataset to compute the Pléiades snow depth accuracy.

The UAV survey and DEMs were performed by a private start-up company at no cost. As we noticed an important spatially heterogeneous bias, we proposed a correction based on a trend assessment. We consider that the UAV dataset remains an interesting, independent, dataset to further assess the *spatial* distribution of the Pléiades HS in spite of its relatively low accuracy. It shows snow depth patterns and transects that are consistent with those retrieved from Pléiades (Fig. 9).

3. Compared to the manual reference data you achieve an underestimation of the HS values of approximately 0.15 m (median). Is there an explanation for this? Could it come from uprising summer vegetation such as bushes that are pressed down to the bottom by the snowpack as reported in Bühler et al. (2016)?

This is an interesting suggestion that we also considered while discussing the results before writing the paper. We associated an error to the snow probe measurements of 0.15 m to take into account this type of uncertainties, but such an error is mostly random and cannot explain the systematic deviation observed in the statistics of the residuals. The z-bias correction of the Pléiades dDEM is characterized by a SD of 0.25 m considering the whole dDEM (see new table 3 above). Therefore, even on a large flat areas as the snow-off football field, it remains difficult to determine a unique b vertical offset and to coregister the summer and winter Pléaides DEMs in z. This can lead to the observed underestimation of the HS values derived from the Pléiades dDEM.

4. You state that the major benefit of the Pléiades sensor is its 12-bit radiometric resolution compared to 11 bits of other comparable satellite sensors. I doubt this statement. I do not think that there is a significant benefit of 12-bit data compared to 11-bit data (while there should be one compared to 8-bit sensors!). Are really all 4096 digital numbers used ?

In my experience also 11-bit data never uses the whole dynamic range.
Could you show some histograms of the input imagery?

We show below the histogram of the winter nadir Pléiades image. The two other histograms from the backward and forwards images are very similar. The number of pixels with saturated values (coded as 4096 in the histogram) represent 3.1 % of the total number of pixels (N=[19 936 246])., Over 95% of the radiometric resolution is actually used.

[Figure]

I would guess that you get very similar results using 11 bit data.

This point is indeed very interesting and could open the door to an intercomparison between DEMs generated over a snow-covered area from various VHR satellite optical sensors with stereo-capability. Such comparison are beyond the scope of the paper but were addressed in the literature. For example, Poli et al. (2015) studied the radiometric and the geometric aspects of the VHR spaceborne imagery from stereo-pairs acquired by WorldView-2 and by GeoEye-1, and a triplet from Pléiades-1A in panchromatic and multispectral mode over a given reference surface. All the data considered showed a good noise robustness and DN stability in both panchromatic and multispectral bands and homogenous values from the Modulation Transfer Function analysis (MTF: this function is used to estimate the spatial performance of an imaging sensor by describing the noise level and the geometrical resolution and sharpness). However, some saturation and spilling effects were observed in GE1 and WV2 images that were not reported from the Pléiades images: "In general the radiometric analysis showed that all the Pléiades images are highly homogeneous, have low noise level and do not present saturation effects".
To acquire an optimal contrast on homogenous snow surface, Buhler et al. 2016 recommend using RAW image storage format with 12 bit, in the case of an UAV acquisition.

With 11/12 bit data you should get good results in shadowed areas but you have to mask them out. Can you explain why you do so ?

The 12 bits encoding does not prevent the decrease of the signal-to-noise ratio in the shaded areas, which decreases the correlation success rate. In addition the Pléiades operator does not allow the user to specify how the sensor gain should be adjusted to the target surface. It remains to be evaluated if it is possible to optimize the sensor gain so that a good correlation is achieved in both illuminated and shaded slopes in a mountainous area.

5. There should be a table listing all available and planned satellite sensors that could potentially be applied for HS mapping including their temporal, spectral, radiometric and spatial resolution.

We propose to include in the supplement this table of VHR optical (civil) satellites that have stereo capabilities comparable to Pléiades, i.e. that could be used for HS mapping based on the same method.

| Satellite platform (launch date) | Stereo-capabiliy | Swath width at nadir | Temporal resolution | Spectral resolution (P) | Radiometric resolution | Spatial resolution at nadir |
|---|---|---|---|---|---|---|
| Pléiades 1A and 1B (2011 and 2012) | tri and stereo | 20 km | 1 day with Pléiades 1A and 1B | 480 - 830 nm | 12 bits | 0.70 m (P) 2.50 m (XS) |
| GeoEye-1 (2008) | stereo | 15.2 km | 8.3 days at 10° and 2.8 days and 28° off nadir look angles, respectively | 450 - 800 nm | 11 bits | 0.46 m (P) 1.84 m(XS) |
| WorldView-1 (2007) | stereo | 17.7 km | 1.7 days at 1 m GSD. 5.4 days at 20° off-nadir (0.52 m GSD) | 400 - 900 nm | 11 bits | 0.50 m (P) |
| WorldView-2 (2009) | stereo | 16.4 km | 1.1 days at 1 m GSD. 3.7 days at 20° off-nadir (0.52 m GSD) | 450 - 800 nm | 11 bits | 0.46 m (P) 1.85 m (XS) |
| WorldView-3 (2014) | stereo | 13.1 km | 1 day at 1 m GSD. 4.5 days at 20° off-nadir or less | 450 - 800 nm | 11 bits | 0.31 m (P) 1.24 m (XS) |
| SPOT 6 and 7 (2012 and 2014) | tri and stereo | 60 km | 1 day with SPOT 6 and SPOT 7 | 450 - 745 nm | 12 bits | 1.50 m (P) 6 m (XS) |

(P) Panchromatic
(XS) Multispectral

6. In my opinion there should be a discussion of potential important applications.
For what applications the identified precision of 0.5 m is sufficient ?
What are the applications where you need better precision for example generated from UAV or laser scanning data?

We agree that the Pléiades snow maps are not accurate enough for a variety of applications. We have added a comment on this aspect in the conclusion:

**This accuracy might be insufficient in areas where the snowpack remains thin even at peak accumulation (North American prairies, semiarid mountains), and for the study of small-scales snow features like sastrugi or penitents**.

However, the potential applications depend from several tradeoffs that snow-product users are prepared to accept. Bühler et al. 2015 provide an overview of the current available methods depicting their strengths and weakness to map snow depth in high alpine terrain at large-scale. We agree with their conclusion: "which method should be applied in a specific case depends on many differents factors and should be evaluated with care". As mentioned in the discussion part, Pléiades derived snow-maps do not present the same accuracies as state-of-the-art airborne Lidar or photogrammetry. It could represent an interesting alternative when such techniques are not available. If the reviewer think it is relevant, wa can add a new table which summarizes some simple considerations of the various techniques to map HS from remote sensing, which columns heading could be:
Remote sensing techniques / (typical) spatial resolution / spatial extent / systematic and random errors in z (HS) / Potential applications.

**Technical corrections:**

P1L1: there is passive microwave; you describe this later in the paper.

P1L1. "To date, there is no direct approach to map snow depth in mountainous areas from spaceborne sensors."
We consider that passive microwave is not a mature approach to map snow depth in mountainous areas as it requires "complex and problematic inversions in order to infer the depth" (Nolan et al., 2015). The kilometer-scale resolution of current passive microwave sensors is not well adapted in *mountainous* areas. Passive microwave sensors offer real-time global SWE estimates but suffer from several problems like subpixel variability in the mountains (Dozier et al., 2016)

Therefore we propose to change the word "direct" by "definitive":
P1L1. "To date, there is no **definitive** approach to map snow depth in mountainous areas from spaceborne sensors."

P1L2: optical stereo satellites

Thank you for that remark. This term will be corrected accordingly in the text:
P1L2: Here,we examine the potential of very-high-resolution (VHR) **optical** stereo satellites to this purpose.

P1L12: please give the calculated precision vs. the UAV data here

According to the reviewer's comment, we completed the sentence as follows:
P1L12: The UAV-derived snow depth map exhibit the same patterns as the Pléiades-derived snow map, **and a median of -0.11 m and a SD of 0.62 m when compared to the snow probe measurements.**

P1L14: I think it is very dangerous to propose the application of remote sensing data without any field data! You need at least some reference measurements to be sure your values are OK. I really suggest deleting this statement!

We totally agree that field data are always welcome to assess remote sensing products! We would always recommend to check the Pléiades results with ground truth observations whenever possible. We only meant that the processing of the Pléiades data does not require mandatory field data like ground control points. We emphasize this aspect because field work can be costly and unsafe in high-elevation mountainous areas. This is the result of a remarkable feature of the Pléiades images (excellent native georeferencing without ground control points).

Not using ground control points before the co-registration of the seasonal DEMs was also emphasized by Nolan et al. 2015 when presenting its airborne photogrammetry-based system to map snow depth:
(abstract section) "The system is simple enough that it can be operated by the pilot without additional assistance and the technique creates directly georeferenced maps without ground control, further reducing overall costs."
(conclusion section) " The mapping technique is based on digital photogrammetry that [...] requires no [...] ground control."

We propose to mitigate the corresponding sentences as follows:
P1L13-14: This study demonstrates the value of VHR stereo satellite imagery to map snow depth in remote mountainous areas **even when no field data are available.**
P16L27-28: Indeed, **the processing of the Pléiades data does not require mandatory field data like ground control points, although such reference measurements are always highly desirable**.

P1L23: From my experience it is more wind, snow avalanches and terrain features that generate the high spatial variability of alpine snow depth distribution.

We modified this sentence as follows:
P1L21--23: Even for small mountain catchments with areas of a few square kilometres, the spatial variability of the snow height and water equivalent is high because **of the elevation gradient of snow fall that is modified by the interaction of snow cover and topography, which leads to a large range of processes: preferential deposition of precipitation, redistribution of snow by wind, sloughing and avalanching (Grunewald,2014).**

P2L27: The main advantage of near-nadir looking instruments against TLS is that you have no holes caused by terrain features such as ridges or bumps and that you can cover the entire area spatially continuous.

Thank you for this comment, we have added this sentence in the manuscript:

**P2L27: However, holes in the dataset caused by convex landforms such as hills or moraines may limit the spatial covering of the TLS acquisition (Buhler, 2016).**

P3L29: UAS?

We considered that the accuracy of the UAV snow depths was not sufficient to extend this analysis to the UAV-Pléiades residuals.

P5L6: It would be nice if some more details on the UAS campaign could be given here. How many images were acquired? What camera did you use? . . ..

In winter, 785 images during four parallels flights were acquired by a Canon IXUS 127 HS mounted on board  (4608 x 3465 pixels, sensor dimension: 6.170 mm x 4.628 mm). The focal length is 4.380 mm, and this value is optimized during the images processing.

During the summer campaign, 964 images during four parallels flights were acquired Sony DSC-WX220 (4896 x 3672 pixels, sensor dimension: 6.170 mm x 4.628 mm). The focal length is 4.572 mm, and this value is optimized during the images processing.

P5L16: approximately 501?

Thank you, we have removed "approximately" from the sentence:

P5L16: We collected up to 501 hand-probed depth measurements on 10 March 2015

P6L13: Why did you choose these spatial resolutions? Please justify.

These are the typical resolutions at which Pléiades DEMs are computed (e.g. Berthier et al. 2014, Marti et al. 2014). Resolutions lower than 1 m are not relevant given the original image resolution and resolutions higher than 4 m will smooth out most of the interesting snow depth features. This was added in the manuscript.

P6L15: Gaussian distribution?

We modified this sentence as follows:
P6L15: "The elevation values at a given grid point were obtained as a weighted average of the elevations of all points in the cloud within the search radius of the grid point, with the Gaussian curve as weighting function..."

P6L17: What are the drawbacks of this method? The winter and summer surface is not similar due to the snow cover. Why is this approach still working? Or did you only use snow free areas to do the SD calculations? Why did you use the 4 m DEMs? This is not clear to me.

This method is frequently used in glaciology (Berthier et al. 2007) and is based on the same principles that Nuth and Kääb 2011. It works because, in our study region of high relief, the average dissimilarity between both DEMs due to the snowpack (as measured by the standard deviation) is lower than the dissimilarity introduced by the horizontal offset across the whole image. So snow-covered area are used for the SD calculation. This method would not perform that well if the analyzed area would present a narrow histogram of aspect (i.e. if the terrain is characterized by a main orientation) and if the terrain was very flat.

We used the 4m DEM because the calculation was faster.

P6L27: Can you give some more details on the classification? What happens if snow is in shadow areas?

Two intensity thresholds were visually adjusted in order to treat specifically the case of the shaded snow surfaces from the general case. Added to the manuscript.

P7L11: I think it is a bit dangerous to sent negative snow depth to no data as you might change the statistics significantly. There might also be many false values, which are slightly positive. These values might differ out more or less. Can you discuss this point and give some indications?

The reviewer is right that it could potentially affect significantly the statistics of the residuals of the comparison between the Pléiades dDEM and the snow probes measurements. However as in indicated in the text (P7-L11) and in the table 3, it concerns only 8 to 10 occurrences (pixels) of the 451 snow probe measurements, therefore 2% or less of the number N of the whole validation dataset.

P7L26: What do you mean by bad stereo orientation? I do also not completely understand you approach with the trend surfaces. These points need a better description.

By trend surface we meant a polynomial interpolation that fits a surface defined by a polynomial function to the input sample points. Here we tried polynomial functions of order 1, 2 and 3. This processing was done using ArcGIS Spatial Analyst toolbox.

P8L9: How do you get to the error value of 0.15 m for probe measurements? Please justify. I would assume it is much less, something around 0.05 m.

We agree that the snow depth can be easily read at 5 cm resolution using a graduated probe. The error due to the probe tilting during the measurement will introduce a few centimeters error. The probe tip penetration in the soil also contributes to increase the error by a few centimeters. The horizontal positioning of the probe sampling point is probably the main source of error. A shift of a few centimeters can change the snow depth by >10 cm because the underlying surface is very heterogeneous. We considered that all these terms represent an error term of 15 cm.

P8L24: (Fig. 9) P10L15: Where and why do you get these data gaps in the point

clouds?

The large data gap in the West of the area corresponds to a lake which serves as a dam for the hydropwer production (visible in figure 2). The other data gaps correspond to the steepest slopes of the watershed. The lake surface was masked as it not considered as a relevant surface (aberrant values). The steepest slopes areas presented a very low correlation rate and led to these data gaps areas.

P1**1**L18: 527.10[**^**]3 what entity is this? Also pts.m2 throughout the document.

P11-L18: This is the number of points (sample size). Added to the manuscript.
P11-L29: The density of the raw photogrammetric point clouds issue from the correlation processes (before rasterization to DEM) are expressed in pts by square meters, or pts.m^2 (please see figure 2 and 3 of the supplement).

P11L19: SD and NAMD are pretty bad compared to the other results. Can you explain why? The NMAD Satellite/Probe is 0.45 m and the NMAD UAV/Probe is 0.35 m. Are they both shifted in the opposite direction? Or how can you get to a NMAD of 0.78 m?

These results are not surprising given the accuracy of each dataset as previously evaluated using the snow probes. In this case we were unlucky as the errors did not compensate.

P12L17: Why?

Considering the influence of the land-cover, the mineral and the shrub classes are associated with the most important dispersion in the residuals distribution (SD are 0.79 and 0.63 m). In the case of the shrub class, this result seems consistent with the fact that the shrubs are highly compressed by the presence of the snowpack.

P14L15: Discuss your [UAV] results in the context of the results published by Harder et al.(2016), Bühler et al. (2016) and Vander Jagt et al. (2015).

We moved the results of Vander Jagt et al. (2015) from the section "Limitation and perspectives" to this section "Comparison to the UAV dDEM" in order to be consistent. The performance presented by Vander Jagt et al. (2015) are very satisfactory, but the snow probe sampling (N=20 snow-probe measurements) limit strongly the significance of the statistics.

To take into account the very recent work in HS mapping by UAV techniques, we rewrote the section "Comparison to the UAV dDEM" incorporating the following considerations:
P14 L15: **Recent works based on UAV systems to map snow depth highlight much better performance than the results reported in this study (2-m-UAV dDEM: SD=0.62m, NMAD=0.35m, median=-0.11m, see Tab. 4). Jagt et al. (2015) used a DSLR camera mounted on a multi-rotor UAV platform to map the snow depth at a very high spatial resolution (GSD 6:10^-3 m) over a small mountainous terrain (0.07 km2) with thick vegetation cover. A comparison with a reduced sample of snow-probe measurements**

(N=20) highlighted an RMSE of 0.096m using GCPs, and 0.184m without (0.084m with one point of coregistration). In Bühler et al. (2016), an UAV-octocopter was used to collect imagery at two alpine sites of the region of Davos in the swiss Alps (1940m and 2500m a.s.l., respectively). The images were acquired with a customized Sony NEX-7 camera with an overlap of 70% along and across-track. Reference data were constituted by plots of one square meter with five manual snow depth measurements. Four snow depth maps were produced and assessed with the manual plots (between 12 and 22 plots according to the map). Accuracies of 0.07 to 0.15m RMSE are reported in a detailed analysis, according to the study sites and the land cover classes. Considering all the reference plots in the valley bottom site, the HS RMSE is 0.25 m and there is an average systematic underestimation of HS by 0.2m. In Harder et al. (2016), a Sensefly Ebee Real Time Kinematic (RTK) UAV was used to collect imagery at a cultivated agricultural Canadian Prairie and a sparsely-vegetated Rocky Mountain alpine ridgetop site (2 300m a.s.l.). In the alpine site, the images were acquired with a Canon IXUS, with a lateral overlap of 85%, a longitudinal overlap of 75%, and a flight altitude of 100m. Multiple acquisitions (43) were performed with careful flight plans. The snow depth was measured with five snow depth measurements in a 0.4m x 0.4m square at the locations of the GNSS survey locations. The average snow depth of the five values was then compared to the snow depth determined by the UAV, with a number of snow depth measurements between three and 20 measurements per flight. The reported snow depth accuracy is characterized by a RMSE of 0.085 m.

In the case of our study, the DEM of the snow-covered area was generated from a unique flight plan. Some problematic flights were reported by Harder et al. (2016) (5 from 43 flights for all sites, or 11.6%) with DEMs showing an RMSE of up to 0.32m. The results mentioned above were extracted from multiple surveys with well spread GCPs and more dedicated survey. We did not use GCPs during the winter survey and only 5 GCPs in summer, not well spread (bottom of the valley only). According to Harder et al. (2016), GCPs are needed to achieve the sub-decimetric accuracy, and a bias correction may also be necessary. Furthermore, residuals of the comparison between the UAV dDEM and the HS manual snow measurements were not filtered (e.g. a statistic criteria like 1 sigma threshold, the land cover classes or the slope). Therefore, despite the discrepancies observed in this study, we consider that the UAV dDEM map was a valuable independent source to evaluate the Pléiades snow depth map because the comparison revealed similar snow depth patterns, while the random and systematic errors of both dDEMs are comparable.

P16L1: Please explain CV

We added its meaning (coefficient of variation):
P16L1 The corresponding **coefficient of variation (**CV**)** value was 0.80 (**CV is** the ratio of the SD to the mean snow depth).

P16L18: Please mention the result of the Satellite / UAS HS comparison

We added the results of the Satellite / UAS HS comparison presented in section 5.3.

P16L28: This statement is dangerous! You need at least hand probe measurements in the file to get an idea about the achieved accuracies.

This sentence was revised (see above).

P16L29: What is the outreach of these results compared to HS measurements with LiDAR, airplanes and UAV? What are potential applications?

See above our response to this comment.

**Figures**

P27 Fig. 5: Please indicate the outline of the UAV extent in the Pleiades extent.

Done.

P28 Fig. 6: The chosen bins are too wide. If you change to a continuous color scale ranging over more than one color, you can make much more details visible. Please adapt the color scale.

We would prefer to keep a discrete color scale. We think that a continuous color scale is not adapted here because the data are skewed. A continuous colormap would highlight the outliers. By using bin color classes we can group the high snow depth values into a single class. The finer distinctions are lost but the map is easier to analyse. To improve the contrast of the snow pattern, we switched to a 7-classes white-blue-purple color scheme from http://colorbrewer2.org.

P31 Fig. 9: Please change the color scale as in Fig. 6.

P31 Fig. 9 (top): To improve the comparison of the snow pattern between the UAV and the Pléiades dérived dDEMs, we used a new sequential color scheme.

P31 Fig. 9: Also the error bins are too wide in my opinion. You only see the very large errors of more than one meter like this.

P31 Fig. 9 (bottom): To improve the readability of the spatial discrepancies between the UAV and the Pléiades dDEMs, we reduced the bins to 0.5 m.

Could you set the profile from one end to the other, like this you do not display the big errors in the northeast because you stop just before that.

We extended the transect from the South-West to the North-East, but in the North-East sector there are lots of "no data" values which are due to the high cliffs.

[Figure]

Legend: Comparison between the UAV and the Pléiades dDEMS.
In black the new extended transect line, in yellow the previous transect line (figure 9).

P32 Fig. 10: How do you get errors compared to the probe measurements for the summer DSM?

The residuals on the summer DEM are computed as follow (equation 6):

$$R_{Z_w} = Z_{SN} - (Z_{w,\mathbf{DGPS}} - HS)$$

(2)

This is explained in the section "Residual analysis on the Pléiades data (4.4)", and the subsection "Residual analysis on the Pléiades data (4.4.2)".

I do not really understand the figure caption, please clarify.

We propose to modify the figure caption as follows:
Figure 10. **Top: Residuals of the comparison between the winter 2 m-Pléiades DEM and the winter DGPS measurements (see equation 5, section 4.4.2), after removal of the bias (median of the residuals).**
**Middle: Residuals of the comparison between the summer 2 m-Pléiades DEM and the estimated summer surface elevation (see equation 6, section 4.4.2), after removal of the bias (median of the residuals).**

**Bottom: Residuals of the comparison between the 2 m-Pléiades dDEM (black bars) and the snow probe measurements according to the probe Id ranked in the ascending HS (red line) order, and after removal of the bias (median of the residuals).**

P36 Tab4: Why is there only one cos value for all snow depth classes ? The same for the slope classes and the aspect classes.

We considered more interesting to present the correlation on the whole dataset to assess the influence of the magnitude of the snow depth in the residuals.

What does the star mean?

Thank you for that remark, we omitted to mention it in that table. We have added::
P36 Tab 4: **Significant correlations (p values <0.05) are marked with asterisks.**

Reviewer #2 : General comments

E. Thibert

Grenoble - 04/03/2016.

General comment

R. Marti and co-authors employ advanced methods of tri-stereo high resolution satellite imagery to retrieve DEMs and snow-cover thickness by DEMs differentiation in a mountainous open terrain. In a rigorous comparison with ground-based manual snow-probing, and UAV-derived DEMs differentiation, the authors find generally good agreement in these comparisons, the discrepancy been explainable by the natural scattering of the data, but also by some residual biases that remained unexplained. These favourable results should confer significant advances and improvements in mapping the snow mantle thickness over large open areas. The paper is almost clear, well organized, and properly focuses the scope of the journal.

We are very grateful to E. Thibert for his attentive revision of our manuscript. We agreed with most of its suggestions as detailed below in the point-by-point response.

*(1) comments from Referees, (2) author's response, (3) author's changes in manuscript.*

The paper would be much more valuable were it also to provide a much clear presentation of the bias corrections in both xy and z-vertical directions for the DEMs derived from Pléiades and UAV flights. The resulting improvements in the residuals before/after adjustments are

not clearly displayed (but dispersed along the text). A summary would be welcome with the numerical values of adjustments as an additional table or a supplementary column in Figure 4.

As suggested by the referee, we added the following table:

**Table 3.** Summary of the differents co-registrations and the bias corrections performed to produde the Pléiades and the UAV DEMs and dDEMS maps. SD means Standard Deviation.

| Input data | Reference data | Type of coregistration | Values of adjustments | Comments |
|---|---|---|---|---|
| 4 m-Pléiades winter DEM | 4 m-Pléiades summer DEM | xy relative coregistration | -5.2 m North +2.8 m East | Workflow data Same shifts applied to the 1 m and 2 m-Pléiades winter DEMs |
| 1 m-Pléiades winter ortho-image | 1 m-Pléiades summer ortho-image | xy relative coregistration | -5.2 m North +3.2 m East | Verification data |
| 1-2-4 m-Pléiades dDEMs | dDEM-snow free footbal field | z relative coregistration | $b_{1\,m} = -0.46\,m$ (SD=0.25 m) $b_{2\,m} = -0.48\,m$ (SD=0.20 m) $b_{4\,m} = -0.44\,m$ (SD=0.15 m) | Workflow data |
| 2 m-Pléiades dDEMs | 78 wide-spread points over snow-free areas | z relative coregistration | Median $b = -0.70\,m$ Mean $b = -0.74\,m$ SD $b = 0.26\,m$ | Verification data |
| 1 m-Pléiades summer ortho-image | 6 wide-spread points on the 0.50 m-IGN ortho-image | xy absolute coregistration | +3 m North (SD=0.38 m) -0.8 m East (SD=0.35 m) | Workflow data Same shifts applied to the dDEMS |
| 0.1 m-UAV-dDEM | 353 wide-spread points over snow-free areas | $\Delta Z$-correction based on a trend surface of order 3 | RMSE: 0.34 m | Post-treatment correction. Same correction applied on the 1 m and 2 m-UAV dDEMs |

Regarding the results of the comparisons between measurement methods, some questions remain:

- Is the remaining bias (-0.12 — -0.16 m) between the Pléiades estimation of snow thickness relative to manual probing significantly different from 0 from your error analysis?

If the reviewer agrees with this approach, we calculated a one-sample t-test on the residuals values to test the null hypothesis that the residuals data comes from a population with mean equal to zero, and this was rejected.

- Same question between Pleiades and UAV DEMs (-0.14 m) ?
A physical explanation should be attempted if you conclude theses biases to be significant.

As stated above, we calculated here a one-sample t-test on the residuals values to test the

null hypothesis that the residuals data comes from a population with mean equal to zero, and this was rejected.

Regarding specifically the UAV image acquisition and process from automatic correlation structure from motion, much more needs to be said about the image orientation, whether the camera model was estimated from a self-calibration in the process, or fixed from standard values or an independent prior calibration.

We agree that the method related to the UAV data was not sufficiently described. However, the UAV-images acquisition, the UAV-image processing, and the UAV-DEMs generation were performed by a private company, and the statistics relative to the photogrammetric processes are not available. Thus, we are limited to comment the orientation residuals in line with the bias identified in the UAV dDEM. However, we now provide additional information, provided by the company, on the acquisition and processing of UAV data: number of images, camera model, focal length and a more detailed description of the method.

The main goal of our study is to evaluate if satellite data can play a role in snow depth mapping. To investigate this question, the 451 snow probe measurements is the foremost validation dataset. We did not use the UAV dataset to compute the Pléiades snow depth accuracy.

A minor point that arises throughout the paper is the indistinct use terms of error in systematic and random meanings, which can sometimes be confusing for the reader. To avoid confusion, I would as much as possible use systematic error, bias, and discrepancy to quantify the incorrectness, and random error and scattering to denote inaccuracy. Moreover, the random error should be defined with respect to the standard deviation (one or 2 times the standard deviation for example). This will help you to decide about the differences between the results to be notified as significant or not.

Thank you for these suggestion. As mentioned below (second substantive comment), maybe a part of that confusion comes from the use of precision and accuracy with a different meaning. As explained below, a strong bias in the residuals may be closely related to a bad accuracy, while a strong scattering in the residuals may be closely related to a bad precision. We agree with the reviewer to refer to the systematic and random errors terms to avoid confusion. We have checked systematically the use of the term "error" in line with this consideration throughout the manuscript and its supplement (also modified accordingly

although not shown here):
P6-L21: "circular error": we refer here to the statistic as employed by Lebegue et al.2010 and by Gleyzes et al. 2013, and therefore we kept it as it.
P8-L8: "random error": use consistent with the reviewer commentary.
P8-L8: "systematic error": use consistent with the reviewer commentary.
P9- L7: "The random error on the DGPS measurements..."
P9-L8: "This term has a random error…"
P9-L9: "..the DGPS error": we deleted the term "error" here as it is redundant.
P9-L9: "Hence, two random error terms exist…"
P9-L10: "more details on the random error calculation".
P11-L9: "systematic error": use consistent with the reviewer commentary.
P11-L11:"random error": use consistent with the reviewer commentary.
P14-L28: "mean error": we replace the term here as follows "a mean of the residuals"

Random errors are combined assuming they are uncorrelated (page 5 of the supplement). This should be stated in the core of the text.

Thank you, this was added in the text:
P9-L7: We assume all the randoms errors to be uncorrelated.

An additional figure would be helpful, setting z-vertical direction and the different almost-horizontal surfaces Zs Zw from Pléiades and UAV and GPS, to define the notations of the variables, their differences, and better understand how standard deviations combine in the equations of the supplement.

To avoid to multiplicate the number of figure, we propose to improve the information available  in the figure 4 (Pléiades worflow) in line with the new table 3 and the equation 1,2, and 3 to help to better understand the different notations in the text.

Here follow some detailed questions, comments, suggestions, and indications of minor typos in the paper.

Substantive comments

P1-L2. Specify ground resolution in meters for consistency with the rest of the paper (all lengths in meter elsewhere)

Thank you for that comment, we corrected the unit accordingly:
P1-L2: Two triplets of 0.70 m-resolution images [...]

P1-L8. To me synonyms are accuracy and precision. Is it meant accuracy (random error) and correctness (bias or systematic error)?

The reviewer arises an important point: we do consider that accuracy and precision have different meaning. Accuracy is how close a measured value is to the actual (true) value. Precision is how close the measured values are to each other. As a bias may be defined as a systematic error which makes all measurements wrong by a certain amount, the precision is unaffected by a bias while accuracy does. In our case, we consider that the accuracy of the snow height (HS) measured by our method is characterized by the median of the residuals distribution. The precision may be assessed through the observed deviation in the residuals distribution characterized by the SD and NMAD values. In that statement, we consider all the HS values as a unique target.

P1-L16-18. I would set at first that snow cover is important in those areas for live and ecosystems, and second for anthropogenic needs.

We modify the beginning of the introduction in line with the reviewer's comment:
P1-L16-18: The seasonal snow cover in mountainous areas sustains mountain glaciers, alters frozen ground through its insulating effect, and plays a major role in mountainous ecosystems and plant survival (Keller et al., 2005). Snow cover is important for hydropower production, irrigation, urban supply, risk assessment and recreation (Barnett et al., 2005).

P1-L20-21. The time for the seasonal snow thickness peak is very dependent on the elevation, varying from December at less than 1000 m a.s.l., to March in the 2000—2700 m range and April-May above 3000 m.

We agree with this comment. In the aforementioned sentence (P1-L20-21), we refer to the snowpack as a "water resource", which implies a persistent snowpack during the favorable accumulation period (November--April). In the Pyrenees, the snow cover is persistent above 1 600 m-- 1 700 m, which correspond to the location of the 0°C isotherm between November--April (e.g. Lopez-Moreno, 2006). Only a very minor part of the Pyrenees are above the 3 000 m. Thus, the time for the seasonal snow thickness peak associated to a persistent snowpack in the Pyrenees is between March-April, although it is strongly affected by the interannual variability (e.g. Lopez-Moreno, 2013). To clarify this point and to take into

account this comment of the reviewer, we propose to add the following sentence:
P1-L21: In the Pyrenees, the accumulation peak associated to the persistent snow pack is generally between March--April (Lopez-Moreno, 2004, 2013).

P1-L23. The natural spatial variability of the snow cover thickness is preliminary due to the variability in precipitations, and post-deposition processes as wind drift, avalanches, snow densification.

To take into account this remark, and the remark of the other reviewer, we modified this sentence as follows:
P1L21--23: Even for small mountain catchments with areas of a few square kilometres, the spatial variability of the snow height and water equivalent is high because of the elevation gradient of snow fall that is modified by the interaction of snow cover and topography, which leads to a large range of processes: preferential deposition of precipitation, redistribution of snow by wind, sloughing and avalanching (Grunewald,2014).

P4-L20. use units in metres for consistency with the rest of the paper

Thank you for that remark, the units were corrected accordingly.

P4-L20. As far as I know, specify that this oversampling is carried out before image delivery

We modified the sentence accordingly:
P4-L20: The Pléiades's pixel depth at acquisition is 12 bits, and the panchromatic images have an initial resolution of 0.70 m, but are oversampled at 0.50 m before image delivery by a post-processing algorithm that was implemented by the French Space Agency (CNES).

P4-L23. specify the local time.

We modified the sentences accordingly:
P4-L23: The snow-free acquisition was programmed on 26 October 2014 (10:53:10, 10:53:31, and 10:53:52 LT).
P4-L30: The second triplet was acquired on 11 March 2015 (10:56:42, 10:57:03, and 10:57:27 LT).

P5-L11—14. It should be clarified between winter and autumn surveys, what set up is used among RTK, ground control points.

In winter there was not GCP apart from the GPS base station, while in autumn there were GCPs. The RTK was activated in both cases.

P5-L23. The z-vertical correctness and accuracy is generally less than that of the xy plane, especially a vertical bias is unavoidable without a geoid model or an independent altimetry adjustment on levelling points.

The accuracy associated to the differential GPS positioning is much probably lower in the z-vertical estimation than in the xy-horizontal estimation, since z-positions are estimated only from positive z-satellite values. However, both accuracies are estimated being lower than 0.1 m after the post-treatment step.  We exported the z-positions estimations as "heights relatives to an ellipsoid" and not relative to the sea level surface, thus a geoid model is unnecessary in that case.

P5-L28. Make it explicit whether your refer to influences in radiometry or ground surface roughness.

Although it could be an interesting possibility, we did not consider here the radiometry. The idea was to interpret the comparison between the Pléiades dDEM and the snow probe measurements according to the land cover classes. To clarify this point, we propose to modify the sentence as follows:
P5-L28: The vegetation types were aggregated into seven classes to reflect the type of land cover that may influence the comparison between the Pléiades dDEM and the snow probe measurements.

P6-L8. Figure 4 introduced before Figure 3 (in page 9 L14)?

Thanks for that remark, we swapped figures 3 and figure 4 accordingly.

P6-L8—11. Define RCP acronym just after "…Earth imagery that uses the RPC…" instead of the next sentence.

Thanks for that remark, we modified the text accordingly.

P6-L13. It should be clarified that each of the three resolution DEMs is retrieved from a

rasterization of the 3 point cloud corpus (otherwise it may be understood that you derive one raster DEM per point cloud).

The description of our method is indeed something ambiguous as we omitted to mention that the three point clouds generated from the three stereoscopic pairs were merged before rasterization. Thus, we modified the text as follows:

P6-L13-14: We generated three point clouds from the three stereoscopic pairs from the *stereo command*, and merged them. The DEMs were rasterized at 1-m, 2-m and 4-m cell sizes from the merged raw point cloud through the *point2dem* command.

P6-L10. It should be clarified by a few words that RCP sets the image-to-ground geometry.

We propose to add the following sentence adapted from the Pléiades imagery user guide (Astrium, 2012):

P6-L10: The RPC model is an analytical model, provided here as meta-data by Airbus Defense and Space (ADS), which gives a relationship between the image coordinates and the ground coordinates with z as the height above an ellipsoid, and which includes both a direct model (image to ground) and an indirect model (ground to image) (Astrium, 2012).

We also modified the method description to be more consistent with the statements of the discussion part (P12 L26-27):

P6-L9: Spatio-triangulation was based on the RPC model which was refined from an automated tie points generation without including ground control points (GCPs).

P6-L15. I would rewrite as: "…grid point, with the Gaussian curve as weighting function…"

Thank you for the suggestion, we modified the text accordingly.

P6-L17. Which is the RMS residual in z after co-registration?

We present in the following table the statistic associated to the z-values before and after the co-registration process.

| Statistics (N=5 944 407) | Before co-registration | After co-registration |
|---|---|---|
| Mean of the difference (m) | -3.43 | -3.3 |
| Median of the difference (m) | -1.51 | -0.91 |
| Standard deviation (m) | 7.33 | 6.91 |
| Normalized Median Absolute Deviation (NMAD) | 5.21 | 3.75 |

We propose to include this table in the supplement if the reviewers or the editor consider it as relevant.

P6-L23. Is it meant that the same xy shift is applied to the higher resolution DEMs without proceeding to a new minimization? Why? And why the 4 m-resolution as the resolution to proceed for all DEMs?

Thank you for this comment. Yes, the same shift was applied to all the DEM (4m-, 2-m and 1-m) without proceeding to a new minimization at finer scale. Since all DEMs were generated from the same raw point cloud (see above), we estimated that the xy-shift should be the same. However, to check this assumption we have run again the minimization algorithm at 2 m and 1 m.

We obtained the following results:

2m DEM:
shift in N/S =   5.23 m
shift in E/W =  -2.50 m
SD =   4.97 m

1m DEM:
shift in N/S =   5.44 m
shift in E/W =  -2.45 m
SD =   4.99 m

As we expected, the best shifts are very similar at 1m, 2m and 4m (we obtained a shift in E/W of -2.83 m and 5.19 m in N/S at 4m). This was noted in the manuscript.

P5-L7—14. Please give more details about the UAV-on-board camera.

It is presumably a non-metric camera. Is it a fixed focal length lens camera? What is the focal length?

We added the following informations to the manuscript:

_in winter, 785 images during four parallels flights were acquired by a Canon IXUS 127 HS (4608 x 3465 pixels, sensor dimension: 6.170 mm x 4.628 mm). The focal length is 4.380 mm;

_during the summer campaign, 964 images during four parallels flights were acquired by a Sony DSC-WX220 (4896 x 3672 pixels, sensor dimension: 6.170 mm x 4.628 mm). The focal length is 4.572 mm.

The focal length is adjusted for each flight during the images processing, as well as the parameters used to modelize the lens distortion.

How on board RTK corrections to tag image centre coordinates and ground control points are used jointly for orientation, and maybe the camera self calibration? I suspect 5 ground control points insufficient for a calibration.

The referee is right that five control points were not enough, especially in this area with steep slopes. If we had to it again we would probably improve the setup of the survey.

P7-L3. Are you really sure that the IGN ortho has a more correct xy referencing than your DGPS ?

We associate an absolute (xy) georeferencing error of 2 m to the IGN orthophoto. Concerning the DGPS position, we consider an absolute xy localization error of 0.1 m. The problem is to interpret the DGPS position in the satellite images. As we did not use reference ground targets easy to photo interpret before the satellite image acquisitions, we are not able to interpret DGPS positions in the satellites images. Therefore, the IGN ortho-image is our unique source to provide an absolute georeferencing before the comparison between the Pléiades dDEM and the  geolocalized snow measurements or the DGPS (x,y,z) coordinates.

P7-L6. Does that bias mean than the co-registration was not optimal? How compares this value with the co-registration residual?

We did not use ground control points (GCPs) during the spatio-triangulation step. Therefore both the summer and the winter DEMs contain a vertical bias as the DEMs are "floating" in (z) and have not been corrected yet. The (xy) co-registration step based on the DEM optimalshift method (P6-L17-20) do not aim at correcting such vertical bias. This is not a 3D-correction method. Therefore, the z-bias correction step is necessary in our workflow, after the xy-co-registration step to provide a final complete 3D co-registration. If the xy-co-registration step present large residuals, the search of a constant vertical bias over the entire image is complicated because it could vary significantly with the slope. However, it should not be affected over large flat areas.

P7-L8. Which photo? Does this refer to the satellite winter and autumn images?

The reviewer is right that the term "photo" is something ambiguous here. We replaced it as suggested:
P7-L8: where b is a constant vertical bias, which is determined from a unique, stable, and flat area of the satellite winter and autumn images that is easy to interpret.

P7-L11. I am not sure it is correct to remove negative snow depths as these values may not be significantly different from zero considering random errors in both you snow probing and dDEM calculations. They may well be acceptable in terms of confidence interval.
Removing some values is nevertheless conceivable considering they might be abnormal (irregular) if they discard negatively from zero at 2 times sigma (or more), according the way you define "aberrant" values.

We removed the negative values because we know that the snow height cannot be negative. This is the last step of the workflow to produce the snow height map. Then we evaluate the error on the snow map. Our primary objective is to evaluate the final product, not to provide a thorough error assessment of Pléiades DEMs or dDEMs. We asses the product that we would generate in an operational framework, or that we would distribute to hydrologists for example.

As answered to the reviewer Y. Bühler, the reviewer E. Thibert is right that it could potentially affect significantly the statistics of the residuals of the comparison between the Pléiades dDEM and the snow probes measurements. However as in indicated in the text (P7-L11) and in the table 3, it concerns only 8 to 10 occurrences (pixels) of the whole 451 snow probe measurements, therefore 2% or less of the number N of the whole validation dataset. We

calculated here below the statistics considering the whole dataset (in bold) to assess the influence of removing the negative snow depths (in normal font) during the Pléiades dDEM assessment:

| dDEM pixel size | Number of snow probing | Median (m) | Standard deviation | NMAD (m) |
|---|---|---|---|---|
| 1 m | 443 / 451 | -0.15 / -0.16 | 0.62 / 0.61 | 0.47 / 0.46 |
| 2 m | 442 / 451 | -0.16 / -0.17 | 0.58 / 0.58 | 0.45 / 0.45 |
| 4m | 441 / 451 | -0.12 / -0.13 | 0.69 / 0.69 | 0.51 / 0.50 |

P7-L15--19. What is the result for the vertical bias calculated from these 78 points. Even if unused later in the paper, how does it compare to the bias calculated from the football field?

This important point is tackled in the section result "5.1. Pléiades and dDEM assessments":
P10-L25: The bias assessment which was performed over the entire Pléiades dDEM (110 km²) and was based on 78 wide-spread values (see section 4.1) indicates a median of - 0.70 m, a mean of - 0.74 m and an SD of 0.26 m. The low SD value and the median difference confirm the possibility to remove a constant bias from a unique area, with 5 small random and systematic errors:
median(football field) - median(entire dDEM) = - 0.22m.

P7-L21—23. A bit more needs to be said about the image orientation (calibration?) process from then UAV acquisition.
Particularly, calibration for non-metric cameras is known to be critical and can generate significant orientation error when processed through automatic correlation Structure-from-Motion'based software as used here.
Which camera model is used for the orientation process? Did you used a simultaneous selfcalibration or a prior calibration? If calibrated, which camera parameters are estimated (decentration, radial distortion and associated polynomial coefficients—how many? , focal length)?

The UAV-data were treated by a private company, excepted the polynomial-based trend correction (cf. table 3). The photogrammetric software used was PIX4D, which uses a prior calibration of the camera model. The focal length and the lens distortion modelling parameters are adjusted for each flight during the photogrammetric treatment by the

software.

We agree that this is a bit frustrating in the mark of a research approach, but the UAV-DEMs are not the main focus of our study, and were used only as comparison data. Recent works focus with a much more complete approach on this technics (Bühler, 2016; Harder, 2016).

Can you give the orientation residuals? This will help you to discuss about the discrepancy between Pléiades and UAV results to be significant or not.

Unfortunately, we do not dispose of such information. The UAV-treatment were performed by a private company (GeoFalco), which did not provide us the orientation residuals. According to the company, the software used (PIX4D) is a kind of "black box" with respect to the orientation residuals values.

P8-L9. It should be explained how probe [random error] is estimated as it is surprisingly high.

Snow probing in mountain area may be challenging and depend in part on the level of experience of the operator: i) an error of 0.05 m may be directly assigned to a reading error related to the graduations ii) it is difficult to maintain the snow probe perfectly vertical (e.g. a 15° inclination of the probe leads to a 0.01 m error at 3.2 m) iii) the vegetation by the snowpack introduce an extra and random error according to the fact that sometimes the bare ground is reached or not. Therefore, we propose to maintain this relative high random error associated to the snow probe sampling.

P8-L16. Instead of error, I would write "…is the median of the dDEM/probe discrepancies.".

The term "median of the errors" is commonly used in the literature to define the second term of the NMAD expression (e.g. (Berthier,2014), (Bühler, 2015)). We propose to use the term "median of the residuals".

P9-L11. Section 4.4.3. From the section 3.4, I expected here an analysis of the ground surface roughness effect, both on manual snow probing and in Zs uncertainty from the 2 DEMs.

A surface roughness index could indeed provide valuable information on the summer DEMs assessment. However, we do not focus on the seasonal DEMs evaluation. We aim at evaluate the DEM difference (dDEM) as it potentially contains the snow height information.

We calculate statistics relative to the comparison between the 2m-Pléiades dDEM and the snow probe measurements for each of the five land cover classes. The idea here is to identify qualitatively a class that might introduce a systematic error (e.g. shrub compression by the snowpack) or a high random error that could be associated to a photogrammetric process issue.

P10-L31. How can you interpret physically this remaining systematic bias for the Pléiades snow thickness estimation? Is it significantly different from zero from your error analysis?

As stated above, we calculated a one-sample t-test on the residuals values to test the null hypothesis and it was rejected.

The z-bias correction of the Pléiades dDEM is characterized by a SD of 0.25 m considering the whole dDEM (see new table 3 above). Therefore, even on a large flat areas as the snow-off football field, it remains difficult to determine a unique b vertical offset and to coregister the summer and winter Pléiades DEMs in z. In that part of the image (football field), the resulting Pléaiades dDEM might be characterized by this systematic offset, which can lead to the observed underestimation of the HS values derived from the Pléiades dDEM.

P11-L5. Why did you force the intercept to be zero and did not fit to Y=aX+b in search of a systematic difference?

We assessed the systematic error through the residuals analysis (table 3). Here, the idea was to estimate how far the Pléiades and the UAV dDEMs are from the HS "signal" including both systematic and random errors. The linear model (y=ax) impedes the compensation of a systematic error instead of an affine model (y=ax+b) that introduces an extra b coefficient. In case of a strong bias, the correlation coefficient is thus affected applying the linear model.

P11-L11. I would expect a much lower value for the residual from the UAV DEM.
Can you comment this in relation to the orientation residual of the images and the overall quality of the DEM geometry?

As stated above, we do not dispose of the orientation residuals and we are not able to comment the relation between the residual from the UAV dDEM and the UAV-images orientation residuals.

P11-L12.The subscript for the residual R is inconsistent with notations from equations 3, 5, 6

?

Thank you for that remark, we corrected the residual expression to be consistent with the definition given by equation 3 in P10-L31 and in P11-L12.

P14-L8—17. I would also question the bias you identified here in relation to the quality of the geometry of the UAV DEM to originate from the orientation of your images.
More needs to be said about cameras, camera models, and the statistics of orientation results.
Instability is frequently associated to inaccurate or residual correlated camera model parameters after least-square adjustment (none unicity of solutions) which can result in poor quality of the geometry of the 3-D model (such as doming or bowl effects) after image orientation.

As stated in the manuscript and above, we are limited to provide valuable informations on the origin of the bias observed in the UAV dDEM, and our discussion remains merely speculative.

In this section, we refer now to recent works on HS mapping by rotor-UAV (Buhler et al. 2016) and winged UAV (Harder et al., 2016).

P14-L27. Is "dynamic" much more appropriated than "resolution" to denote the 12-bits depth?

The term "dynamic" seems indeed more appropriate, and we modified the sentence accordingly. However the expression "radiometric resolution" seems also correct and commonly employed in the literature (eg. Lee et al. 2008 (p.832) cited in the manuscript).

P14-L28. To make a better distinction between accuracy/correctness concepts, use bias instead of "error" when you refer to a systematic error. Or systematically qualify errors as random or systematic to avoid confusion.

Please consider the general answer above. Here, we cite the results of a publication (Lee et al. 2008).
P14-L28: "mean error": we replace the term here as follows "a mean of the residuals"

P15-L18. Inflect somewhat writing "…for clear-sky/limited cloud cover conditions…"

We modified the sentence as suggested:
P15-L18: As for all optical sensors, the main drawback of the Pléiades constellation is the need for clear-sky or with limited cloud cover conditions to obtain suitable images

P15-L34. "In hydrology and water resource applications, there remains…"

We completed the sentence as suggested.

P16-L8. This result is not trivial. The snow cover thickness you can estimate is not significantly different from that of the snowpack model, considering the overall uncertainty in both estimations.

As the snow cover thickness we estimated from Pléiades data and the snow cover thickness simulated by the snowpack model do not correspond to the same hydrological cycle, 2014-2015 and 2011-2012 respectively, we did not go further in that comparison.

P16-L26—29. I would mitigate/inflect your conclusions here mentioning that you nevertheless need an altimetry control/adjustment on a snow-free flat surface —as your z-vertical bias corrections demonstrate— that you have to infer on each satellite imagery. But this control surface can be located kilometres apart and at lower elevations.

The reviewer is right that several aspects of the conclusion should be mitigated here. We modified the text as follows:
P16-L26--29: Indeed, the processing of the Pléiades data does not require mandatory field data like ground control points, although such reference measurements are always highly desirable. An adjustment on a snow-free flat surface, which can be located kilometres apart and at lower elevations, is needed to correct a vertical bias in the Pléiades DEMs difference.

Supplement

It is not clear to me why probe appears in equations 3, 4 and 5. I would only expect this term in the uncertainty associated to the comparison of dDEMs with HS.

In equation 4 and 5, we propagate the error sources identified in equation 3. Snow probe error is present in the equation 3 because we evaluate the Pléiade and the UAV-snow free

DEM from a Z_summer estimation based on the DGPS Z_winter value minus the snow probe height.

Figures

P23-Figure 1 — caption. "Pyrénées mountains. Bottom: Bassiès…"
Identically, in the legends of top right and main maps : "Bassiès"

Thank you for that remark, we corrected "Bassies" by "Bassiès" accordingly. "Pyrénées" is generally written "Pyrenees" in the english literature, so we would prefer keeping its english orthography.

P24-Figure 2 — caption. "Comparison of terrestrial oblique pictures taken by automatic cameras…"
The local time is mentioned here but not in the text.

Thank you for that remark, now the local time has been also mentioned in the text (please see above). We also added the word "terrestrial" as proposed by the reviewer.

P30-Figure 8 —left map. Add a title "Snow depth" at bottom right corner as you did for the 2 other maps at top left.

Thank you for that remark, the title "Snow depth" was added to the left map.

P32-Figure 10 — caption. Add plot colours "…and the 2m-Pléiades dDEM (black bars) according to the probe Id ranked in the ascending HS (red line) order (see equation section 4)."

Thank you for that remark, we added the "black bars" and "red line" in the caption.

P32-Figure 10 — labels. Point of notation: define the units for the residual errors as (m), and not (in m). Same for HS in the right-hand axis.

Thank you for that remark, we now define the units by (m) and not anymore by (in m).

Stylistic comments

P1-L1. At present…

As both expression seem equivalent in english, we would prefer to keep "To date".

P1-L19. Snow Covered Area…. Snow Height

Thank you for that remark, we added uppercases in the acronym definition.

P1-L20. Snow Water Equivalent (SWE)

Thank you for that remark, we added uppercases in the acronym definition.

P4-L1. Bassiès and Pyrénées

As Bassiès has no english equivalent word in the english language, we would keep it in its french orthography. "Pyrénées" may be written "Pyrenees" in the english literature, so we would prefer keeping its english orthography.

P4-L7 " 6.6°C and the mean annual precipitation is 1640 mm

Thank you for that remark, we completed the sentence accordingly.

P4-L9 "…and 25% by vegetation-free rock and bare soils."

Thank you for that remark, we modified the sentence accordingly.

P4-L30. erase "which was"

We erased "which was" from the sentence.

P4-L30. "…115km², and centred on the Bassiès catchments, as achieved for snow-free images."

Thank you for that remark, we completed the sentence accordingly.

P4-L32. "…-14° along track direction…"

Thank you for that remark, we corrected this grammatical aspect.

P5-L1. "…-6.4° across track direction."

Thank you for that remark, we corrected this grammatical aspect.

P5-L9 "…mean Ground Sampling Distance (GSD)…".

Thank you, we added uppercases in the acronym definition.

P5-L13. "…installed in a nearby mountain refuge…"

We modified this sentence by providing some more information:
P5-L13 [...] GPS-base, which was installed on the flat dropping zone of the mountain refuge during the survey.

P5-L16. "We collected up to 501 hand-probed…"

Thank you for that remark, we modified the text accordingly.

P5-L16. "…10 March 2015, at the time of the UAV survey…"

Thank you for that remark, we modified the text accordingly.

P5-L19. "…snow probes with lengths of 2.2m and 3.2m,…"

Thank you for that remark, we corrected this grammatical aspect.

P6-L3. "…to i) limit the areas potentially masked by the rugged topography…"

Thank you for that remark, we modified the text accordingly.

P6-L24. "…1-m resolution from their respective DEMs…"

Thank you for that remark, we corrected this typographical error.

P7-L24. "…snow probe"

Thank you for that remark, we corrected this typographical error.

P8-L1. "…the flat dropping zone of the mountain hut…"

Thank you for that remark, we modified "heliport" by "flat dropping zone".

P8-L15. Here and in lines 5 and 18 on page 11, don't know the correct sign to use for multiplication between dot and cross signs following the journal style ? my preference is cross…

It seems that several convention coexist to indicate the multiplication operator: dot (e.g. (Berthier, 2014)), or no sign at all (e.g. (Grünewald, 2010) or (Buhler,2015)). We propose to put no sign between variables to indicate a multiplication operator.

P9-L14. space to add between 3.2 m and (Fig.3).

Thank you for that remark, we corrected this typographical error.

P12-L27. "tie-point measurements, whose effect is equivalent…"

Thank you for that remark, we corrected this grammatical aspect.

P14-L4. space to erase after "safe"

Thank you for that remark, we corrected this typographical error.

P14-L28. In upper case letters "Motion Unit"

Thank you for that remark, we corrected the sentence accordingly.

P15-L3. Unclear sentence to correct, it seems that "by" is missing before Jagt et al. (2015)?

Thank you for that remark, we corrected the sentence accordingly:
P15-L3: A DSLR camera that was mounted on a UAV platform was used over a small mountainous terrain (0.07 km²) with thick vegetation cover by Jagt et al. (2015) to map the snow depth [...]

P35-Table 3 — caption. Remind the reader what the acronym NMAD denotes.

Thank you for that remark, we added the NMAD acronym definition in table 3 (p.35) and table 4 (p.36).

P36-Table 4 — caption. It would be helpful to remind the reader that * denotes significant

correlations as mentioned in Table 3 caption.

Thank you for that remark, we omitted to mention it in that table. We added the same sentence as in table 3:
P36 Tab 4: Significant correlations (p values <0.05) are marked with asterisks.

P36-Table 4. What does reversed brackets denote in the interval bin column?

Reverse brackets are used to exclude the limit of the interval bin, which allow to consider the values of the given variable (col.1 of table 4) only one time in the statistics calculation (col. 4-6).

[revised manuscript text omitted]

- in winter, on the 10 March 2015, 785 images during four parallels flights with 70% lateral and longitudinal overlaps were acquired by a Canon IXUS 127 HS camera (4608 x 3465 pixels, sensor dimension: 6.170 mm x 4.628 mm, focal length: 4.380 mm);

- in summer, on the 13 July 2015, 964 images during four parallels flights with 70% lateral and longitudinal overlaps were acquired by a Sony DSC-WX220 camera (4896 x 3672 pixels, sensor dimension: 6.170 mm x 4.628 mm, focal length: 4.572 mm).

The flight altitude was maintained at approximately 150 m, which provided a mean  Ground Sampling Distance (GSD) from 0.10 to 0.40 m.  Both the winter and snow-free acquisitions were achieved under very clear sky conditions. On-board RTK corrections were performed at 20 frequency. UAV-orientation was improved during the winter survey through the use of a GPS-base, which was installed  on the flat dropping zone of the mountain refuge during the survey. Five georeferenced ground targets were placed in the valley bottom during the summer, and identified on  the UAV-images to improve the absolute positioning accuracy.

**3.3 Snow probing**

We collected  up to 501 hand-probed depth measurements on 10 March 2015,  at the time of the UAV survey, and one day before the Pléiades acquisition (Tab. 1). Because of the limited available time on the field, we

attempted to cover an area that could represent of a large part of the catchment topography. The distance between each sample ranged from 10 to 30 m. We used two types of snow probes with  lengths of 2.2 m and 3.2 m, respectively. The snow probing coordinates were recorded by using a differential GPS (DGPS) with a mean of 15 acquisitions (one per second) per probe location. We used the Trimble Geo XH 2008 (GPS) and Geo XH 6000 (GPS and Glonass). Post-treatment corrections were collected from a base that was 21 km away, specifically the French "Réseau Géodésique Permanent" network (RGP, base: "Mercus-Garrabet"). This process enabled us to achieve 0.1-m accuracy in the horizontal and vertical directions of the snow probing locations.

**3.4 Land cover map**

A 2008 land-cover map, which was updated by a field survey in July 2015, was generated through an object-based approach and expert-interpretation of aerial photographs (Sheeren et al., 2012; Houet et al., 2015) (see the supplement for the land cover map). The vegetation types were aggregated into seven classes to reflect the type of land cover that may influence the comparison between the Pléiades dDEM and the snow probe measurements: mineral surfaces (bare soil and rocks), water surfaces (rivers and lakes), peatland, low grass (rangeland, grassland, and subalpine meadows), shrubs, trees (conifer and deciduous), and unknown.

**4 Methods**

**4.1 Production of DEMs, orthoimages and dDEMs from Pléiades images**

A tri-stereoscopic acquisition was considered to i) limit the areas  potentially masked by the rugged topography of the studied catchment, ii) improve the correlation by providing different B/H ratios, and iii) obtain a nearly nadir image to improve the ortho-rectification process and accuracy of the absolute co-registration offset.

Snow-free and winter Pléiades DEMs were generated from the image triplets through the Ames Stereo Pipeline (ASP, version 2.4.8.), an open source automated stereogrammetry software by NASA (Broxton and Edwards, 2008; Moratto and Broxton, 2010; Willis et al., 2015) (Fig. 3). The ASP was primarily designed to create DEMs of ice and bare-rock surfaces. The ASP supports any Earth imagery that uses the  Rational Polynomial Coefficients ( RPC) camera model format. The RPC model is an analytical model, provided here as meta-data by Airbus Defense and Space (ADS),  which gives a relationship between the image coordinates and the ground coordinates with z as the height above an ellipsoid, and which includes both a direct model (image to ground) and an indirect model (ground to image) (ASTRIUM, 2012). Spatio-triangulation was based on the RPC model which was refined from an automated tie points generation without including ground control points (GCPs). We parameterized the ASP to project the images into an epipolar geometry to reduce the search range before the correlation (Normalized Cross Correlation) and triangulation steps. We generated three point clouds from the three stereoscopic pairs from the *stereo* command, and merged them. The DEMs were rasterized at 1-m, 2-m and 4-m cell sizes from the

 merged point cloud through the *point2dem* command. Resolutions lower than 1 m are not relevant given the original image resolution and resolutions higher than 4 m will smooth out most of the interesting snow depth features. The elevation values at a given grid point were obtained as a weighted average of the elevations of all points in the cloud within the search radius of the grid point, with the  Gaussian curve as weighting function (see the supplement for

5   the ASP's parameters) (NASA, 2015).

Four-meter snow-free and winter DEMs were horizontally co-registered by iteratively shifting the winter DEM with respect to the summer DEM (reference) by minimizing the standard deviation (SD) of the elevation difference distribution (Berthier et al., 2007). The final horizontal shifts were applied to the winter DEM were: $-5.2$ m in northing and $+2.8$ m in easting. We obtained similar results by computing the optimal shift at 1 m and 2 m resolution. This result is consistent with the expected

10  localization precision that was provided by the RPCs from the Pléiades images. Without ground control points (GCPs), the horizontal location accuracy of the images was estimated at 8.5 m for a circular error at a confidence level of 90% (CE90) for Pléiades-1A and 4.5 m for Pléiades-1B (Lebegue et al., 2010; Gleyzes et al., 2013). The same shift was applied to the 2-m and 1-m winter DEMs.

Winter and snow-free nadir images were rectified at 1-m resolution  from their respective DEMs, before co-registration.

15  By picking 6 wide-spread corresponding points on the snow-free and winter images, the mean shifts were: $-5.2$ m in northing (SD=0.7 m) and $+3.2$ m in easting (SD=0.5 m), which are consistent with shifts from the DEM co-registration technique. The low SD values indicate that the horizontal shift was almost constant in the image. A classification of the image pixels into snow and snow-free classes based on intensity thresholds was performed on the winter ortho-image (Tab. 2). Two intensity thresholds were visually adjusted in order to treat specifically the case of the shaded snow surfaces from the general case.

20  dDEMs were produced at 1-m, 2-m and 4-m spatial resolution by subtracting the snow-free DEM from the winter DEM on a pixel by pixel basis:

$$\Delta Z_0 = Z_w - Z_s \tag{1}$$

where $Z_w$ is the pixel value in the winter DEM, and $Z_s$ is the pixel value is the snow-free DEM.

An absolute horizontal shift in the Pléiades DEMs was estimated from six wide-spread points that were identified on an

25  aerial orthophoto from ©IGN("Institut National de l'Information Géographique et Forestière"), which presents an absolute accuracy of approximately 2 m. The shift between the snow-free Pléiades ortho-image and the IGN ortho-photo was: $+3$ m (SD=0.38 m) in northing and $-0.8$ m (SD=0.35 m) in easting. The dDEMs were then shifted based on this absolute horizontal offset, to be consistent with the DGPS and the georeferenced snow-probe measurements.

Then, we removed a constant vertical bias from $\Delta Z_0$ (Eq. 1) to obtain the final dDEMs:

30  $$\Delta Z = \Delta Z_0 - b \tag{2}$$

where $b$ is a constant vertical bias, which is determined from a unique, stable, and flat area of the  satellite winter and autumn images that is easy to interpret. We chose to evaluate $b$ from a snow-free football field in the image that was 5 km from the mountain refuge (Fig. 1). The value of $b$ was assumed to be equal to the median of the dDEM distribution on the football

field. After this bias correction, dDEM pixels with negative values were classified as "no data", which include 8 to 10 pixels that correspond to a snow probe measurement (Tab. 4). We classified the percentage of negative dDEM pixel values over the Bassiès catchment according to the presence of snow, and excluded shadow areas from steep rocks or cliffs.

Verifying whether a vertical bias that is measured over a small portion of a dDEM at low elevations (football field) can be
5  used to correct an entire dDEM is very important. To test this assumption, we extracted 78 wide-spread values from the 2-m-Pléiades dDEM before bias correction (Eq. 1). We photo-interpreted these points on snow-free rock areas, roads or bare soil in the absolute geo-referenced winter ortho-image by avoiding the steepest slopes ($< 30°$), and by covering a large elevation range ($790 - 2510$ m). We did not use this information to remove the bias because we aimed to evaluate a simple workflow that could become operational.

10  ## 4.2 Production of UAV DEMs and dDEMs from the UAV images

UAV DEMs were generated from the overlapping drone images (70% end lap, 70% side lap) by using the ©PIX4D software, which uses a structure-from-motion (SfM) algorithm (Westoby et al., 2012). The focal length as well as the lens distortion modeling parameters of the cameras were adjusted for each flight during the automatic PIX4D workflow. Five GCPs were available in summer to improve the snow-free images orientation. Except the position of the GPS-base, no GCPs were available
15  during the winter survey, thus the winter images were co-registered to the summer images to improve their orientation. Generated point clouds were rasterized at 0.1-m, 1-m and 2-m cell sizes for both the snow-free and winter DEMs. Subsequently, 0.1-m, 1-m and 2-m-dDEMs were obtained by differencing the corresponding snow-free DEM from the winter DEM. The UAV-images acquisition, the UAV-image processing, and the UAV-DEMs generation were performed by a private company (Tab 1).

20  After an initial comparison with the  snow probe measurements, a marked planar bias–oriented SW-NE was identified on the dDEMs. Comparing the winter UAV DEM values to the winter DGPS measurements (N=343) showed that the bias resulted from a bad stereo orientation, which led to some deformations in the winter DEM. To correct that bias, we extracted 353 wide-spread values from the 0.1-m-UAV dDEM at locations where the snow depth was supposed to be zero based on the winter ortho-image (emerging bare rock). We generated trend surfaces of order 1, 2 and 3 based on these values, and subtracted
25  them from the 0.1 m-dDEM. The trend surface are generated from a polynomial interpolation that fits a surface defined by a polynomial function to the input sample points. Here we tried polynomial functions of order 1, 2 and 3. This processing was done using ArcGIS Spatial Analyst toolbox. 
[revised manuscript text omitted]
 potentially less stable than UAVs with rotors  (Bühler et al., 2016), although recent works have highlighted their great potential for snow mapping in high-alpine catchments  even in relative windy conditions (Harder et al., 2016). We noted large mismatches between the Pléiades and UAV dDEM maps for steep slopes, which could be due to incorrect flight plans or, lens calibration, co-registration errors (James and Robson, 2014).

Recent works based on UAV systems to map snow depth highlight much better performance than the results reported in this study (2-m UAV dDEM: SD=0.62 m, NMAD=0.35 m, median=-0.11 m, see Tab. 4). Jagt et al. (2015) used a DSLR camera mounted on a multi-rotor UAV platform to map the snow depth at a very high spatial resolution (GSD $6.10^{-3}$ m) over a small mountainous terrain (0.07 km²) with thick vegetation cover. A comparison with a reduced sample of snow-probe measurements (N=20) highlighted an RMSE of 0.096 m using GCPs, and 0.184 m without (0.084 m with one point of co-registration). In Bühler et al. (2016), an UAV-octocopter was used to collect imagery at two alpine sites of the region of Davos in the swiss Alps (1940 m and 2500 m a.s.l., respectively). The images were acquired with a customized Sony NEX-7 camera with an overlap of 70% along and across-track. Reference data were constituted by plots of one square meter with five manual snow depth measurements. Four snow depth maps were produced and assessed with the manual plots (between 12 and 22 plots according to the map). Accuracies of 0.07 to 0.15 m RMSE are reported in a detailed analysis, according to the study sites and the land cover classes. Considering all the reference plots in the valley bottom site, the HS RMSE is 0.25 m and there is

an average systematic underestimation of HS by 0.20 m. In Harder et al. (2016), a Sensefly Ebee Real Time Kinematic (RTK) UAV was used to collect imagery at a cultivated agricultural Canadian Prairie and a sparsely-vegetated Rocky Mountain alpine ridgetop site (2 300 m a.s.l.). In the alpine site, the images were acquired with a Canon IXUS, with a lateral overlap of 85%, a longitudinal overlap of 75%, and a flight altitude of 100 m. Multiple acquisitions (43) were performed with careful flight plans.

5     The snow depth was measured with five snow depth measurements in a 0.4 m x 0.4 m square at the locations of the GNSS survey locations. The average snow depth of the five values was then compared to the snow depth determined by the UAV, with a number of snow depth measurements between three and 20 measurements per flight. The reported snow depth accuracy is characterized by a RMSE of 0.085 m.

     In the case of our study, the DEM of the snow-covered area was generated from a unique flight plan. Some problematic flights

10 were reported by Harder et al. (2016) (5 from 43 flights for all sites, or 11.6%) with DEMs showing an RMSE of up to 0.32 m. The results mentioned above were extracted from multiple surveys with well spread GCPs and more dedicated survey. We did not use GCPs during the winter survey and only 5 GCPs in summer, not well spread (bottom of the valley only). According to Harder et al. (2016), GCPs are needed to achieve the sub-decimetric accuracy, and a bias correction may also be necessary. Furthermore, residuals of the comparison between the UAV dDEM and the HS manual snow measurements were not filtered

15 (e.g. a statistic criteria like 1 $\sigma$ threshold, the land cover classes or the slope). Therefore, despite the discrepancies observed in this study, we consider that the UAV dDEM map was a valuable independent source to evaluate the Pléiades snow depth map because the comparison revealed similar snow depth patterns, while the random and systematic errors of both dDEMs are comparable.

**6.3.1**

20 ## 6.4   Limitations and perspectives

[revised manuscript text omitted]

20  values and the corresponding snow probe measurements present a median of $-0.16\,\mathrm{m}$ and an SD of $0.58\,\mathrm{m}$. An independent dDEM map was generated through a winged UAV photogrammetric survey on the same date based on a similar workflow. Despite some outliers, the UAV dDEM map was also successfully validated by the snow probe measurements (median of the residuals is $-0.11\,\mathrm{m}$, SD is $0.62\,\mathrm{m}$). The comparison between the 2-m-Pléiades and the 2-m-UAV dDEMs is characterized by a relatively scattered distribution of the residuals mainly due to some outliers in the UAV dDEM: median is $-0.14\,\mathrm{m}$ (mean

25  is $-0.06\,\mathrm{m}$), SD is $1.47\,\mathrm{m}$, and NMAD is $0.78\,\mathrm{m}$. The snow cover features that were obtained by Pléiades DEM differencing were consistent with those that were derived from the UAV acquisition. The correlation between the snow heights from both techniques is statistically significant, even if some discrepancies were present on the steepest slopes.

The accuracy might be insufficient in areas where the snowpack remains thin even at peak accumulation (North American prairies, semiarid mountains), and for the study of small-scales snow features like sastrugi or penitents. Further studies should

30  focus on influences of the snow height, the topography and the land cover on the accuracy of Pléiades-derived snow heights based on Lidar-derived snow height maps. Our validation dataset limited the analysis to gentle slopes or relatively flat areas and snow heights up to $3.2\,\mathrm{m}$. The shadows that are projected onto slopes create a lack of radiometric contrast in both snow-free

and winter images and constitute an inherent limitation to optical sensors. Other limitations include obstructions by the forest canopy and cloud cover.

These results are promising because they open the possibility to retrieve the snow height at a metric horizontal resolution in remote mountainous areas that are difficult to access. Indeed,  the processing of the Pléiades data does not require  mandatory field data like ground control points, although such reference measurements are always highly desirable. An adjustment on a snow-free flat surface, which can be located kilometres apart and at lower elevations, is needed to correct a vertical bias in the Pléiades DEMs difference. The size of the study area could vary from several square kilometres to several hundreds of square kilometres.

*Acknowledgements.* The Pléiades images acquisition was supported by the CNES through the ISIS Pléiades programme (Projet CRYOPYR–4500107601). This work was supported by the Région Midi-Pyrénées and the University of Toulouse through the CRYOPYR project. We thank the OHM Vicdessos Human-Environnemental Observatory and the Labex DRIIHM (*Dispositif de Recherche Interdisciplinaire sur les Interactions Hommes-Milieux*) for logistic and financial support. We also warmly thank Audrey Chone, Yoann Malbeteau, Yoann Moreau, and Vincent Rivalland who helped us to collect the snow-probe measurements. The authors deeply acknowledge Michael J. Willis for his help in ASP parameterization to produce the DEMs. We also thank Joaquin Maria Munoz Cobo Belart for its useful comments that improved the manuscript.

[revised manuscript text omitted]

**Figure 10.**  Top: residuals of the  comparison between the 2 m-Pléiades winter DEM and the winter DGPS measurements (see equation 5, section 4.4.2), after removal of the bias (median of the residuals). Middle: residuals of the comparison between the snow-free 2 m-Pléiades DEM  and the estimated summer surface elevation (see equation 6, section 4.4.2), after removal of the bias (median of the residuals). Bottom: residuals of the comparison between the 2 m-Pléiades dDEM (black bars) and the snow probe measurements according to the probe Id ranked in the ascending HS (red line) order (see equation 3 section 4.3), and after removal of the bias (median of the residuals).

**Table 1.** Data sources and description. ADS means "Airbus Defence and Space". GEODE and CESBIO are both laboratories of the Toulouse University (France). ©GeoFalco is a French start-up specialized in UAV data acquisition and processing.

| Data sources | Acquisition date | Institution (acquired by) | Ground Sampling Distance (m) | Photogrammetric information | Products (resolution) |
|---|---|---|---|---|---|
| 1B-Pléiades triplet | 26 Oct. 2014 | ADS | 0.70 – 0.73 m | B/H=0.22; 0.23; 0.45 | snow-free DEM (1 m;2 m; 4 m) |
| Snow probe measurements | 10 March 2015 | GEODE CESBIO | 10 – 30 m | - | validation dataset |
| UAV photographs | 10 March 2015 | GeoFalco | 0.10 – 0.40 m | 70% end-lap, 70% side lap | winter DEM (0.1 m;2 m) |
| 1A-Pléiades triplet | 11 March 2015 | CNES | 0.70 – 0.73 m | B/H=0.22; 0.26; 0.48 | winter DEM (1 m;2 m; 4 m) |
| UAV photographs | 13 Jul. 2015 | GeoFalco | 0.10 – 0.40 m | 70% end-lap, 70% side lap | snow-free DEM (0.1 m;2 m) |

**Table 2.** Percentage of potential outliers and no data in the dDEM values, considering the catchment area, the snow-covered area of the catchment, and the snow-covered area of the catchment located out of the shadows due to the high cliffs (called here below "sunny snow").

| Data source | Pixel size | No data | Percentage of | | | | |
| --- | --- | --- | --- | --- | --- | --- | --- |
| | | | $\Delta Z < 0\,\mathrm{m}$ | | | $\Delta Z > 15\,\mathrm{m}$ | |
| | | | in the catchment | on snow | on sunny snow | in the catchment | on snow |
| | 1 m | 2.4 % | 22.4 % | 14.7 % | 9.4 % | 0.14 % | 0.09 % |
| Pléiades tri-stereo | 2 m | 1.7 % | 24.5 % | 17 % | 11.3 % | 0.15 % | 0.1 % |
| | 4 m | 1.2 % | 22 % | 14.5 % | 9.8 % | 0.17 % | 0.1 % |

**Table 3.**  Summary of the differents co-registrations and bias corrections performed to produce the Pléiades and the UAV  DEMs and dDEMS maps. SD means Standard Deviation. The term worflow metrics refer to the  data presented in the figure 3.

| Input data | Reference data | Type of coregistration | Values of adjustments | Comments |
|---|---|---|---|---|
| 4 m-Pléiades winter DEM | 4 m-Pléiades summer DEM | xy relative coregistration $\Delta X$rel.$\Delta Y$rel. | -5.2 m North +2.8 m East | Workflow metrics (same shifts applied to the 1 m and 2 m-Pléiades winter DEMs) |
| 1 m-Pléiades winter ortho-image | 1 m-Pléiades summer ortho-image | xy relative coregistration | -5.2 m North (SD=0.70 m) +3.2 m East (SD=0.50 m) | Verification metrics |
| 1-2-4 m-Pléiades dDEMs | dDEM-snow free football field | z relative coregistration $b$ | $b_{1\,m} = -0.46$ m (SD=0.25 m) $b_{2\,m} = -0.48$ m (SD=0.20 m) $b_{4\,m} = -0.44$ m (SD=0.15 m) | Workflow metrics |
| 2 m-Pléiades dDEMs | 78 wide-spread points over snow-free areas | z relative coregistration $b$ | Median $b = -0.70$ m Mean $b = -0.74$ m SD $b = 0.26$ m | Verification metrics |
| 1 m-Pléiades summer ortho-image | 6 wide-spread points on the 0.50 m-IGN ortho-image | xy absolute coregistration $\Delta X$abs.$\Delta Y$abs. | +3 m North (SD=0.38 m) -0.8 m East (SD=0.35 m) | Workflow metrics (same shifts applied to all the Pléiades dDEMS) |
| 0.1 m-UAV-dDEM | 353 wide-spread points over snow-free areas | $\Delta Z$-correction based on a trend surface of order 3 | RMSE: 0.34 m | Post-treatment correction. Same correction applied on the 1 m and 2 m-UAV dDEMs |

**Table 4.** Statistics relative to the comparison between the Pléiades and the UAV dDEMs to the snow probe measurements, according to the pixel resolution. Significant correlations (p values <0.05) are marked with asterisks. NMAD means Normalized Median Absolute Deviation (Höhle and Höhle, 2009).

| Data source | dDEM pixel size | Number of snow-probe sampling | Median (m) | Standard deviation (m) | NMAD (m) | Spearman correlation $cor_s(\Delta Z, HS)$ |
|---|---|---|---|---|---|---|
| Pléiades tri-stereo | 1 m | 443 | -0.15 | 0.62 | 0.47 | 0.71* |
| | 2 m | 442 | -0.16 | 0.58 | 0.45 | 0.72* |
| | 4 m | 441 | -0.12 | 0.69 | 0.51 | 0.67* |
| Pléiades front/nadir stereo pair | 4 m | 411 | -0.54 | 0.64 | 0.53 | 0.62* |
| Pléiades nadir/back stereo pair | 4 m | 450 | 0.13 | 0.61 | 0.47 | 0.73* |
| UAV photographs | 0.1 m | 343 | -0.07 | 0.63 | 0.38 | 0.8* |
| | 1 m | 336 | -0.15 | 0.62 | 0.36 | 0.79* |
| | 2 m | 339 | -0.11 | 0.62 | 0.35 | 0.79* |

**Table 5.** Statistics relative to the comparison between the 2m-Pléiades dDEM (tri-stereo) and the snow probe measurements, according to the snow depth, slope and aspect, and the land cover classes. Significant correlations (p values <0.05) are marked with asterisks. NMAD means Normalized Median Absolute Deviation (Höhle and Höhle, 2009).

| Variable | Interval bins | Number of snow-probe sampling | Median (m) | Standard deviation (m) | NMAD (m) | Spearman correlation $cor_s(\lvert R_{\Delta Z}\rvert, HS)$ |
|---|---|---|---|---|---|---|
| Snow depth | [0 ; 0.5 m] | 25 | 0.24 | 0.22 | 0.31 | |
| | ]0.5 m; 1 m] | 65 | -0.01 | 0.46 | 0.33 | |
| | ]1 m; 1.5 m] | 94 | -0.07 | 0.44 | 0.39 | 0.3* |
| | ]1.5 m; 2 m] | 114 | -0.24 | 0.60 | 0.34 | |
| | ]2 m;2.5 m] | 72 | -0.32 | 0.68 | 0.54 | |
| | ]2.5 m; 3.2 m] | 46 | -0.63 | 0.56 | 0.39 | |
| Slope | ]0°; 5°] | 150 | -0.10 | 0.42 | 0.32 | |
| | ]5°; 10°] | 117 | -0.19 | 0.53 | 0.41 | |
| | ]10°; 15°] | 81 | -0.30 | 0.53 | 0.59 | 0.26* |
| | ]15°; 20°] | 63 | -0.30 | 0.79 | 0.7 | |
| | > 20° | 31 | -0.18 | 0.93 | 0.75 | |
| Aspect | North | 159 | -0.20 | 0.6 | 0.43 | |
| | East | 113 | -0.15 | 0.63 | 0.48 | - |
| | South | 134 | -0.16 | 0.55 | 0.46 | |
| | West | 43 | -0.12 | 0.55 | 0.39 | |
| Land cover | All classes | 442 | -0.16 | 0.58 | 0.47 | 0.72 |
| | Mineral | 56 | -0.2 | 0.79 | 0.60 | 0.74 |
| | Water | 21 | -0.32 | 0.55 | 0.50 | 0.67 |
| | Low grass | 140 | -0.16 | 0.49 | 0.35 | 0.74 |
| | Shrub | 140 | -0.15 | 0.63 | 0.51 | 0.68 |
| | Peatland | 84 | -0.15 | 0.51 | 0.42 | 0.69 |